# Translational reprogramming in response to accumulating stressors ensures critical threshold levels of Hsp90 for mammalian life

Kaushik Bhattacharya[1], Samarpan Maiti[1], Szabolcs Zahoran[2], Lorenz Weidenauer[3], Dina Hany [1], Diana Wider [1], Lilia Bernasconi[1], Manfredo Quadroni [3], Martine Collart [2] & Didier Picard [1] ✉

The cytosolic molecular chaperone Hsp90 is essential for eukaryotic life. Although reduced Hsp90 levels correlate with aging, it was unknown whether eukaryotic cells and organisms can tune the basal Hsp90 levels to alleviate physiologically accumulated stress. We have investigated whether and how mice adapt to the deletion of three out of four alleles of the two genes encoding cytosolic Hsp90, with one Hsp90β allele being the only remaining one. While the vast majority of such mouse embryos die during gestation, survivors apparently manage to increase their Hsp90β protein to at least wild-type levels. Our studies reveal an internal ribosome entry site in the 5′ untranslated region of the Hsp90β mRNA allowing translational reprogramming to compensate for the genetic loss of *Hsp90* alleles and in response to stress. We find that the minimum amount of total Hsp90 required to support viability of mammalian cells and organisms is 50–70% of what is normally there. Those that fail to maintain a threshold level are subject to accelerated senescence, proteostatic collapse, and ultimately death. Therefore, considering that Hsp90 levels can be reduced ≥100-fold in the unicellular budding yeast, critical threshold levels of Hsp90 have markedly increased during eukaryotic evolution.

Stress is inevitable. Every organism is repeatedly exposed to stress, either intrinsic or extrinsic[1]. Stress stimuli (stressors) induce cell-autonomous (cellular) or non-autonomous (organismal) stress responses, which may intersect and functionally interact[2–6]. In an organism, exposure to stress has two different outcomes, adaptation or failure to adapt; the latter would be indicative of stress susceptibility[7,8]. Whereas "good adaptation" is advantageous for natural selection and developmental robustness[9], "bad adaptation" drives diseases like cancer[10]. Alternatively, hypersensitivity of diseased cells to therapeutic drugs can constitute a "good susceptibility" for the benefit of patients[11–14]; "bad susceptibility" causes age-related

degenerative processes, including neurodegeneration and aging itself[15–17]. How a living organism manages its cellular and organismal stresses determines its fate.

To confront cellular stress, organisms express stress sensors and managers, such as molecular chaperones, which can act as both[17]. Molecular chaperones, including Hsp70 and Hsp90, are evolutionarily conserved proteins responsible for the assisted protein-folding processes of native, misfolded, or structurally labile proteins[18–21]. Intriguingly, during evolution from prokaryotes to eukaryotes, while overall proteome complexity dramatically increased without any accompanying gain of genes for new types of core molecular chaperones, a

[1]Department of Molecular and Cellular Biology, University of Geneva, Geneva, Switzerland. [2]Department of Microbiology and Molecular Medicine, University of Geneva, Geneva, Switzerland. [3]Protein Analysis Facility, Center for Integrative Genomics, University of Lausanne, Lausanne, Switzerland. ✉e-mail: didier.picard@unige.ch

plethora of co-chaperones appeared[22]. Molecular chaperone functions in eukaryotic organisms may be critically regulated and controlled by co-chaperones in a context-specific manner, and these molecular chaperone machines might be associated with several fundamental biological processes beyond protein folding/refolding. These include transcription[23], translation[24–26], protein translocation[27], and protein degradation via the proteasome and chaperone-mediated autophagy[28–31].

In contrast to the situation in prokaryotes, Hsp90 is essential for the viability and growth of eukaryotic cells and organisms even under normal permissive conditions[17,29,32]. Budding yeast, a lower eukaryote, can grow normally with as little as 5% of its total Hsp90 protein levels at a slightly reduced temperature[33]; with even more severely reduced levels, it can still grow albeit with a significant growth retardation[34]. Hsp90, together with its co-chaperones and other molecular chaperones, is an integral part of the system maintaining cellular protein homeostasis (proteostasis). Not surprisingly, failure to maintain proteostasis is associated with developmental failure, neurodegeneration, and premature aging[3,16,17].

Similar to yeast, mammals also have two different cytosolic Hsp90 isoforms, Hsp90α (encoded by the gene *HSP90AA1* in humans; *HSP82* in yeast) and Hsp90β (encoded by *HSP90AB1* in humans; *HSC82* in yeast), a stress-inducible and a constitutively expressed isoform, respectively[35,36]. Individually, either of them is dispensable in yeast[37] and human cancer cell lines[38]. Hsp90α and Hsp90β share extensive sequence identity, and largely but not completely overlapping molecular and cellular functions[35,36]. In the mouse, the absence of Hsp90α primarily causes male sterility[39] and retinal degeneration[40], whereas the absence of Hsp90β causes early embryonic lethality[41].

Although the two cytosolic Hsp90 isoforms are potentially differentially required in mammals, how and to which extent low levels of total Hsp90 can be tolerated by mammals was unknown. Furthermore, it was unclear whether and how mammals can adapt to the genetic or pharmacological loss of Hsp90 by activating compensatory mechanisms, and what physiological states would actuate these mechanisms.

Here, we report our investigation of these questions at the cellular and organismal levels using different Hsp90 mutant mouse and cellular models under normal physiological and stressed conditions. We find that mammalian life requires much higher threshold levels of the molecular chaperone Hsp90 than budding yeast, that mammalian cells can fine-tune the expression levels of Hsp90 in response to physiologically accumulating cellular stress, and that an internal ribosome entry site (IRES) in the 5′-UTR of the *Hsp90ab1* mRNA reprograms translation in stressed conditions contributing to maintaining threshold levels of Hsp90. These mechanisms serve to promote stress adaptation and survival of the organism.

## Results

### Reduced frequency of live mice with a single *Hsp90* allele at birth indicates distorted segregation

Wild-type (WT) mice have two Hsp90α-encoding alleles (*Hsp90aa1*) and two Hsp90β-encoding alleles (*Hsp90ab1*). To investigate the impact of reducing the number of *Hsp90* alleles, we set up crosses to generate mice with a compound genotype of homozygous *Hsp90aa1* and heterozygous *Hsp90ab1* knockouts (herein referred to as 90αKO 90βHET), along with 90αKO[39], 90βHET, 90αHET 90βHET, and WT mice (Supplementary Fig. 1a, b). We observed a striking reduction in the frequency of viable 90αKO 90βHET mice at birth relative to the expected Mendelian inheritance for the cross between 90αHET 90βHET and 90αKO mice (Fig. 1a, Supplementary Fig. 1c). To evaluate whether the reduced frequency of viable 90αKO 90βHET mice is due to embryonic lethality, the frequency of 90αKO 90βHET embryos was determined at embryonic stages E13.5 and E8.5 using the same breeding strategy (Supplementary Fig. 1c). We found a gradual loss of 90αKO 90βHET embryos during gestation (Fig. 1a). The heterozygous

deletion of *Hsp90ab1* has the most severe effect on the viability at birth in combination with a homozygous *Hsp90aa1* knockout (Fig. 1b, Supplementary Fig. 1c–g). Very rarely, developmentally retarded and morphologically deformed dead 90αKO 90βHET pups were born from the breeding of 90αHET 90βHET with 90αKO mice (Fig. 1c). These findings lead us to speculate that the loss of 90αKO 90βHET embryos may start as early as at implantation and continue until birth. We further confirmed the reduced frequency of 90αKO 90βHET mice at birth by backcrossing 90αHET 90βHET male and 90αKO 90βHET female littermates; the results of this experiment further demonstrated that the ability of embryos to survive is not genetically or epigenetically transmissible to the offspring by adult survivors with the 90αKO 90βHET (Fig. 1d, Supplementary Fig. 1g). Remarkably, the few 90αKO 90βHET pups that are born alive seem to thrive normally with a lifespan (811.4 ± 152 days; *n* = 5) similar to that of other Hsp90 mutants and corresponding to that of WT mice reported in the literature[42].

### Increased Hsp90β protein levels correlate with survival of 90αKO 90βHET mice

To investigate the molecular basis of why a small proportion of 90αKO 90βHET pups manage to survive, we performed quantitative label-free proteomic analyses of brain, liver, and muscle. The choice of these three tissues was based on the differential ratios between Hsp90α and Hsp90β protein levels (Supplementary Fig. 2a). We included one set each of male and female mice in the proteomic analysis to avoid any sex-specific differences between the genotypes. As a quality control, we performed a correlation analysis with the data of the two independent replicates of each tissue. The calculated Pearson correlation coefficients (*r*) for all comparable datasets were close to 1, indicative of highly correlated replicates (Supplementary Fig. 2b). We considered changes significant when the comparison of the averages of these sets for a given protein indicated a Log2 fold change of >0.4 or <−0.4 with a *p*-value of <0.1.

The proteomic analyses indicated that only a minor proportion of the identified proteins of brain, liver, and muscle tissues are differentially expressed between the genotypes (Supplementary Fig. 3a–c, Supplementary Data 1). Overall proteostasis appears to be maintained across all genotypes, including in the 90αKO 90βHET survivors. Next, we focused on the Hsp70-Hsp90-related chaperones, co-chaperones, and other stress-responsive proteins from the proteomic dataset of brain. Remarkably, compared to 90βHET mice, the loss of one *Hsp90ab1* allele in the absence of both *Hsp90aa1* alleles (90αKO 90βHET mice) is significantly compensated by the overexpression of Hsp90β from the remaining allele (Fig. 2a, Supplementary Fig. 4a, Supplementary Data 1). This phenomenon is also apparent across multiple other organs in 90αKO 90βHET mice, notably lungs, eyes, kidney, spleen, and heart (Supplementary Fig. 4a). Based on the relative levels of the two isoforms in wild-type tissues, and assuming that both alleles of a given gene contribute equally, one can calculate how much Hsp90 protein should still be made by the remaining allele(s) without any rectification. In tissues with the 90αKO 90βHET genotype, the expected Hsp90 levels range from about 25% in brain to about 40% in the muscle. What is experimentally observed is a rectification of Hsp90β levels that is correlated with the expected loss of total Hsp90 across the analyzed tissues, rather than with the loss of alleles for this or that isoform (Fig. 2b, Supplementary Fig. 4b, Supplementary Data 1). Mouse adult fibroblasts (MAFs) established from ear biopsies of WT and Hsp90 mutant mice displayed a similar pattern with regards to Hsp90β expression (Fig. 2c). Although obvious in MAFs, an increase of Hsp90α levels in 90βHET mouse tissues is not consistently seen across the two sexes and the analyzed tissues (Fig. 2b, c, Supplementary Fig. 4a, Supplementary Data 1). Further analysis of the tissue extracts of 90αHET 90βHET mice revealed that Hsp90β levels are more consistently increased to WT levels compared to those of Hsp90α upon the loss of one allele in each of the two *Hsp90* isoforms in the same

individual (Supplementary Fig. 4c). Moreover, though lacking one allele of *Hsp90ab1*, Hsp90β protein levels in cells and tissues with the 90αKO 90βHET genotype are indistinguishable from those associated with the 90αKO genotype (Fig. 2c, d, Supplementary Figs. 4a and 5a). Remarkably, the rectification of the Hsp90β protein levels associated with the 90αKO 90βHET genotype pushes them at least up to the WT levels and sometimes even surpasses that, notably in the brain, eye, heart, and lung tissues (Fig. 2d and Supplementary Figs. 4a and 5a). Considering that Hsp90β is the constitutively expressed and only poorly stress-inducible Hsp90 isoform[35,36], this increase of the Hsp90β level is unexpected.

To understand the molecular basis of the increase of the Hsp90β levels, we quantitated the *Hsp90* mRNA levels. Comparisons between the transcript and protein levels across the relevant genotypes revealed that raised mRNA levels are not a major contributor to the increase of the Hsp90β protein levels in 90αKO 90βHET mice (Fig. 2c,

d and Supplementary Figs. 4a, and 5a, b). While this is clearly the case for a whole panel of tissues, we cannot rule out that transcriptional reprogramming of the Hsp90β gene *Hsp90ab1* is involved in augmenting the Hsp90β protein levels in tissues and cell types that we did not check. We tentatively concluded from these experiments that a translational or post-translational mechanism may increase Hsp90β protein levels to ensure that total Hsp90 protein levels are maintained above a critical threshold level, as is most evident for the 90αKO 90βHET mice that survive to adulthood.

## Hsp90β-specific translational activation in 90αKO 90βHET cells and tissues

Since we found that the increase of Hsp90β levels in 90αKO 90βHET cells and tissues may be translational, we checked the translation rate of Hsp90β in WT and mutant MAFs. Analysis of the puromycin incorporation into nascent polypeptides revealed a reduced

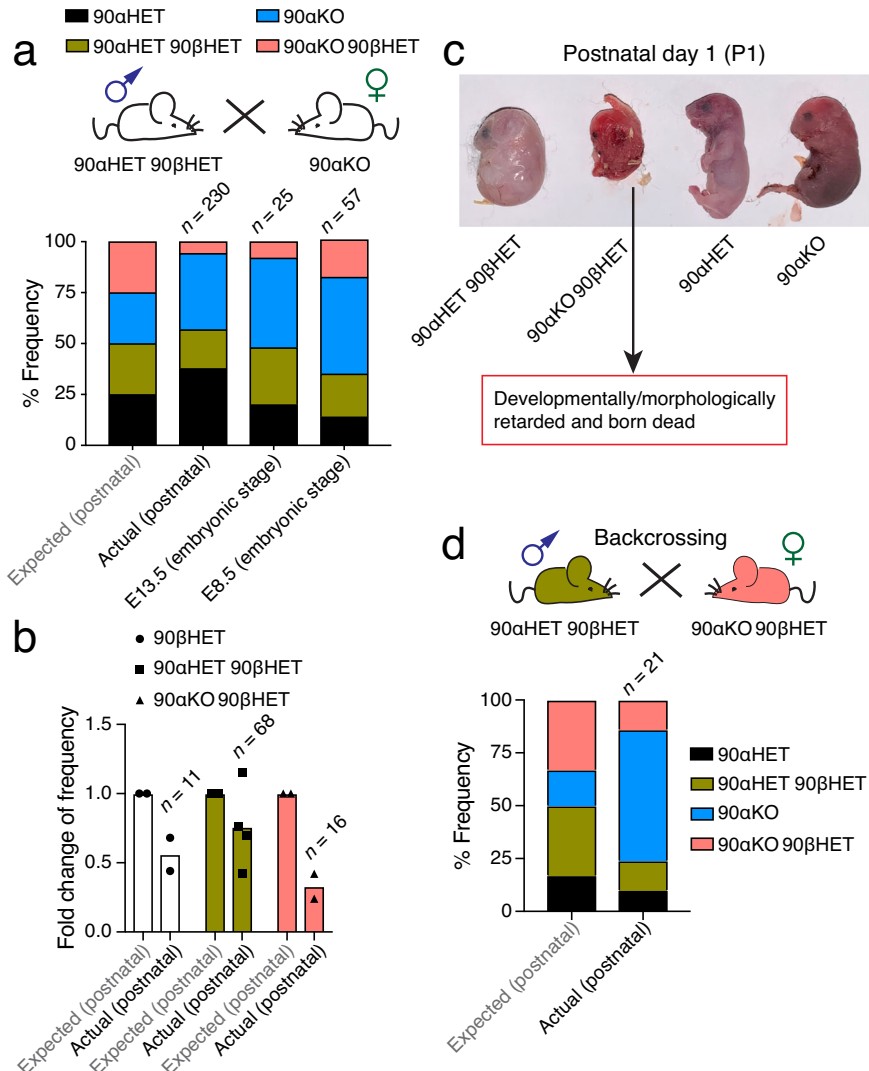

**Fig. 1 | Segregation distortion in mice carrying variable numbers of *Hsp90* alleles. a** Expected Mendelian inheritance and observed (actual) segregation in live pups (postnatal) or embryos (E8.5 and E13.5) of the indicated genotypes from crossing 90αHET 90βHET and 90αKO mice. Only female 90αKO mice were used in this breeding strategy since male mice of the identical genotype are sterile[39]. **b** Fold change of the expected and observed genotype frequencies for live pups resulting from different breeding strategies illustrated in Supplementary Fig. 1c–g. The expected frequency of a particular genotype from a specific breeding was set to 1. Each data point corresponds to a specific breeding strategy that produced

offspring of the indicated genotype with the number *n* standing for the total number of pups with that genotype. **c** Morphology of newborn pups at postnatal day 1 (P1) from crossing 90αHET 90βHET and 90αKO mice. The 90αKO 90βHET pup is stillborn and developmentally retarded in this set. **d** Expected and observed genotype frequencies in live pups (postnatal) from backcrossing 90αHET 90βHET male and 90αKO 90βHET female. Only female 90αKO 90βHET mice were used in this breeding strategy. n indicates the total number of analyzed pups or embryos. Source data are provided as a Source Data file.

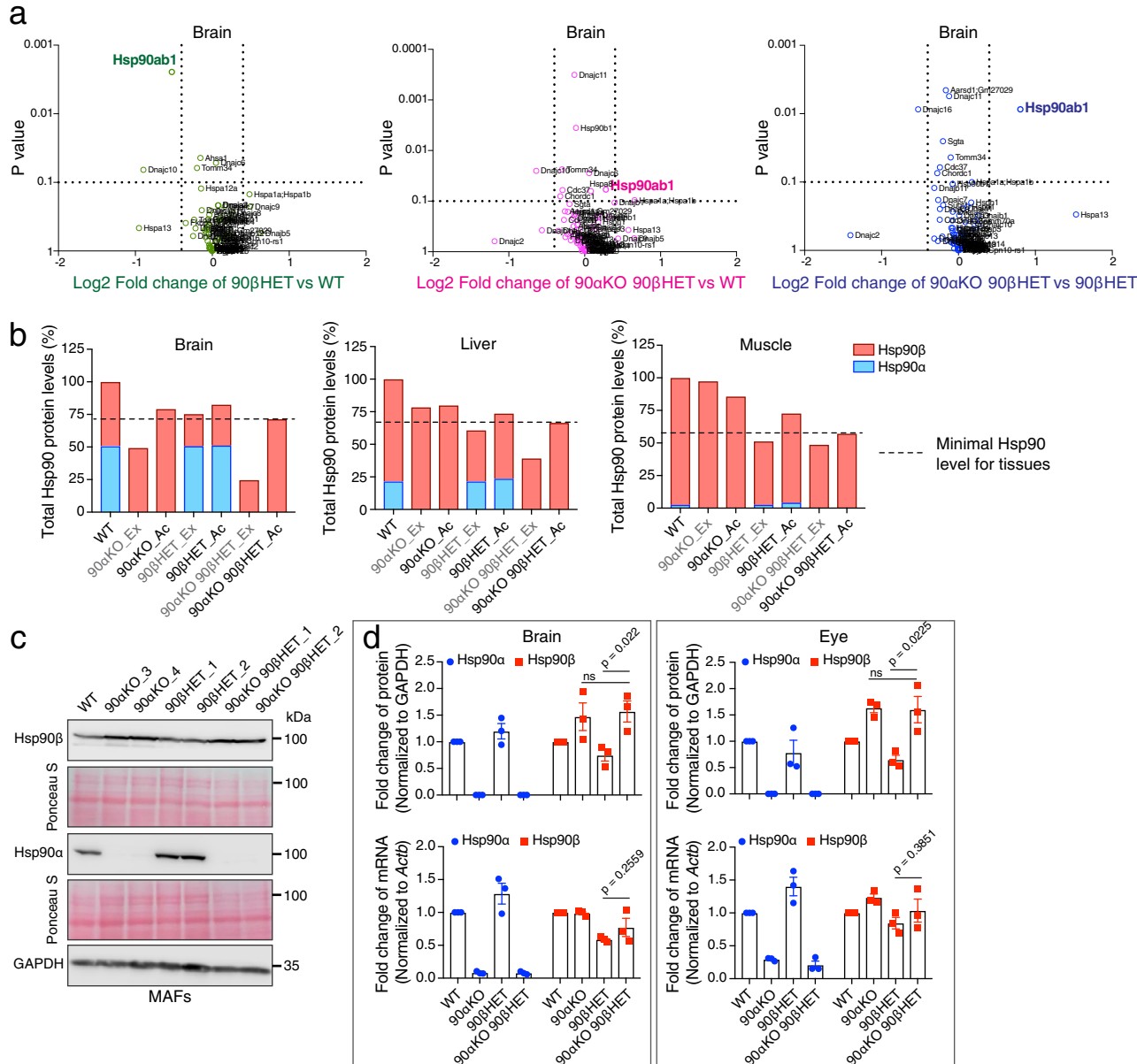

**Fig. 2 | Increased Hsp90β protein levels in 90αKO 90βHET survivor mice.**
**a** Volcano plot of the normalized fold changes of the Hsp70-Hsp-related chaperones, co-chaperones, and other stress-responsive proteins determined by quantitative label-free proteomic analysis of brain ($n = 2$ biologically independent samples). Log2 fold change of >0.4 or <−0.4 with a $p$-value of <0.1 were considered to be significant differences for a particular protein. The comparisons between samples of different genotypes are shown in different colors. **b** Total Hsp90 protein levels and relative proportions of the Hsp90α and Hsp90β isoforms (measured by quantitative label-free proteomic analysis) for the indicated tissues of animals of the different genotypes. Ex, expected Hsp90 protein levels (these are mathematically derived and based on the observed *Hsp90* allele-specific contributions to total Hsp90); Ac, actual (observed) Hsp90 protein levels (derived from proteomic analyses). Total Hsp90 protein levels in WT samples were set to

100%. Dotted lines indicate the experimentally observed minimum Hsp90 protein levels for the indicated mouse tissues (means of $n = 2$ biologically independent samples). **c**, Immunoblots of Hsp90α and Hsp90β from MAFs (representative of $n = 4$ biologically independent experiments). Two independent clones of Hsp90 mutant MAFs were analyzed. The Ponceau S-stained nitrocellulose filters and GAPDH immunoblot signal serve as loading controls. **d** Bar graphs represent normalized protein and mRNA expression of Hsp90α and Hsp90β isoforms in mouse brain and eyes of animals with the indicated genotypes. WT protein or mRNA levels were set to 1 ($n = 3$ biologically independent samples). The data are represented as mean values ± SEM for the bar graphs. The statistical significance between the groups was analyzed by two-tailed unpaired Student's $t$-tests. ns non-significant $p$-values, MAFs mouse adult fibroblasts. Source data are provided as a Source Data file.

global translation in 90αKO 90βHET MAFs (Fig. 3a). Whether the increase in global translation in 90αKO cells can be confirmed with other MAFs and cell types remain to be seen, but we note that it is not the case in brain (see below). What is noteworthy is that the relative translation rate of Hsp90β in 90αKO 90βHET compared to WT and Hsp90 mutants MAFs, including 90αKO MAFs, is *increased*, ensuring equivalent steady-state levels of Hsp90β protein (Fig. 3a, b). Consistent with this finding, the abundance of ribosome-associated *Hsp90ab1*

mRNA is increased in 90αKO 90βHET compared to 90βHET MAFs (Fig. 3c), although there is no difference in the total *Hsp90ab1* mRNA level (Supplementary Fig. 6a). Despite the fact that 90αKO 90βHET MAFs lack one *Hsp90ab1* allele and display a reduced global translation, the abundance of ribosome-associated *Hsp90ab1* mRNA over inputs in 90αKO 90βHET MAFs is indistinguishable compared to WT or 90αKO MAFs (Fig. 3c). This further supports the notion of an Hsp90β-specific translational activation in 90αKO 90βHET cells

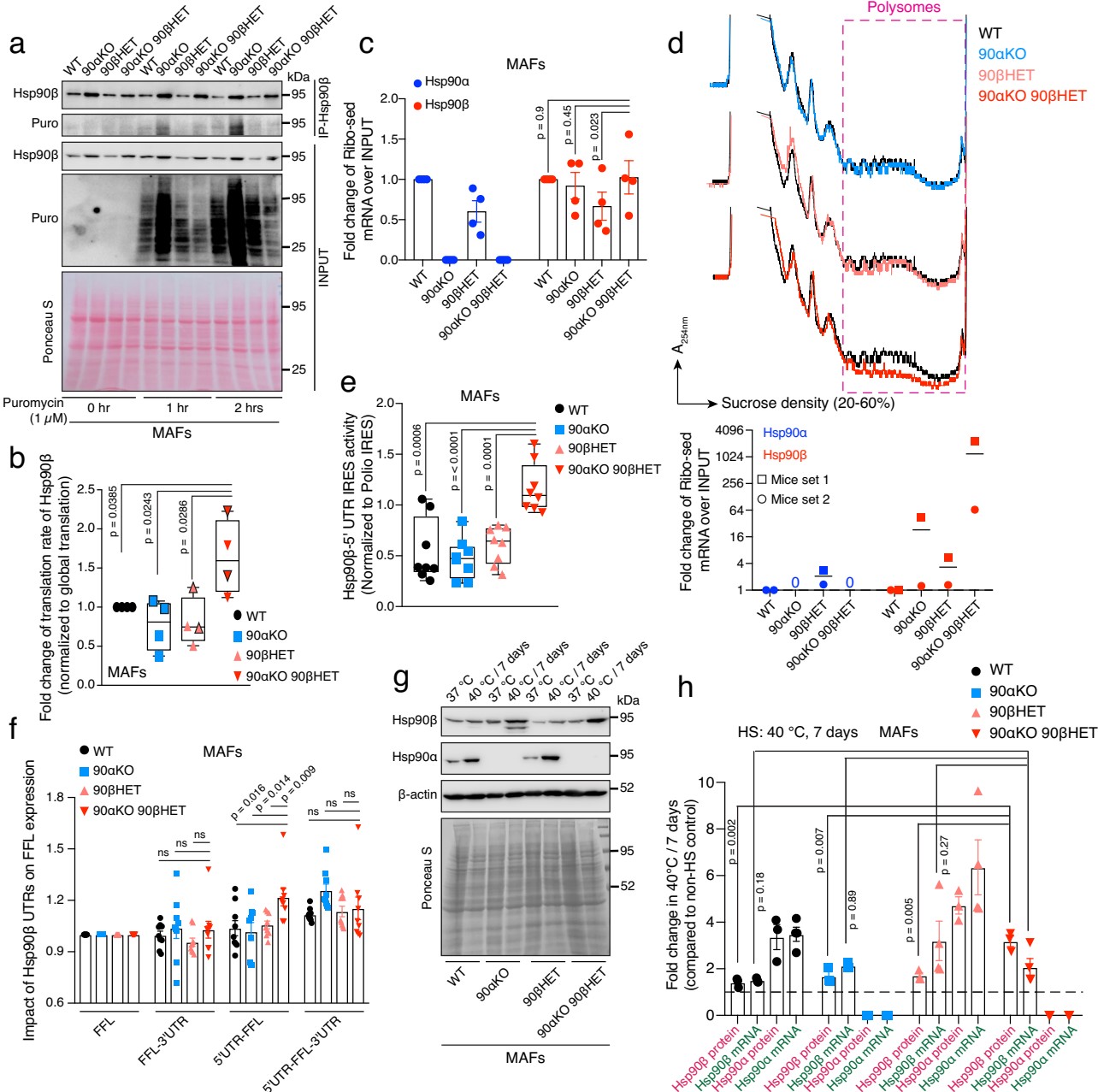

**Fig. 3 | Increased translation of the *Hsp90ab1* mRNA in 90αKO 90βHET is IRES-driven. a** Translation rate of Hsp90β based on the incorporation of puromycin (Puro). The Ponceau S-stained nitrocellulose filter serves as a loading control for inputs (*n* = 2 biologically independent experiments). **b** Relative translation rate of Hsp90β compared to the global protein translation rate in the MAFs of indicated genotypes. Densitometric scores were calculated from the experiments of Fig. 3a. Puromycin labeling for 1 and 2 h, each with two replicates (data points without and with outlines, respectively), was plotted together for a particular genotype. Hsp90β translation in WT samples was set to 1. Note that, solely as an indication, the *p*-values were calculated as if there were four replicates per genotype, and that a similar trend is seen when the time points are plotted separately. **c** Normalized fold change of ribosome-associated Hsp90α and Hsp90β mRNAs over inputs in MAFs of the indicated genotypes (see also Supplementary Fig. 6a; *n* = 4 biologically independent samples). *Actb* was used as a reference gene. **d** Representative polysome profiles of mouse brain (set 1) of the indicated genotype. Each profile from a Hsp90 mutant mouse brain was compared to that of WT (Top). The graph below the polysome profiles shows the normalized fold change of brain polysome-associated Hsp90α and Hsp90β mRNAs over inputs (bottom; *n* = 2 biologically independent

mice sets). *Gapdh* was used as a reference gene. **e** IRES activities of the 5′-UTR of mouse Hsp90β mRNA, normalized to the respective poliovirus IRES activities (*n* = 8 biologically independent samples; see also Supplementary Fig. 7b for details on the bicistronic reporter plasmids); values >0 indicate IRES activity. **f** Impact of the UTRs of mouse Hsp90β mRNA on the translation of firefly luciferase (FFL) (*n* = 9 biologically independent samples; see also Supplementary Fig. 7e for details on the FFL reporter plasmids). Activities of the FFL reporter without UTRs were set to 1. Note that the *Y*-axis starts at 0.6. **g** Representative immunoblots of Hsp90α and Hsp90β from MAFs after 7 days in culture at 37 °C and 40 °C (*n* = 3 biologically independent experiments). β-actin and the Ponceau S-stained nitrocellulose membrane serve as loading controls. **h** Normalized fold change of mRNA (green labels, *n* = 4 biologically independent samples) and protein (pink labels, *n* = 3 biologically independent samples) expression of Hsp90α and Hsp90β from MAFs after 7 days in culture at 40 °C. Expression values at 37 °C were set to 1 (dashed line). The bar graphs show mean values ± SEM. Box plots with whiskers show the data distribution from minima to maxima, and the lines across the boxes indicate the median values. The statistical significance between the groups was analyzed by two-tailed unpaired Student's *t*-tests. Source data are provided as a Source Data file.

(Fig. 3c, Supplementary Fig. 6a). Polysome profiling of brain samples of 90αKO 90βHET mice confirmed the reduced global translation and a remarkably enhanced abundance of the polysome-associated *Hsp90ab1* mRNA, which conceivably maintains the total Hsp90 protein at WT levels in the brain (Fig. 3d, Supplementary Fig. 6b, also see Fig. 2b).

To evaluate the formal possibility that the translational upregulation of Hsp90β associated with the 90αKO 90βHET genotype may be due to a global stress response, ribosome association of mRNA for other molecular chaperones and co-chaperones was checked. Except for *Hsp90ab1* mRNA, no other tested mRNA is noticeably enriched in the ribosome-bound fraction from 90αKO 90βHET MAFs compared to 90βHET MAFs (Supplementary Fig. 6c). This finding is paralleled by the insignificant upregulation of the protein levels of other molecular chaperones and co-chaperones in 90αKO 90βHET compared to 90βHET MAFs and mouse tissues (Fig. 2a (right), Supplementary Fig. 6d–f, Supplementary Data 1). Therefore, the translational upregulation of Hsp90β in 90αKO 90βHET cells and tissues is a specific response that occurs as a result of the loss of three quarters of the alleles and the specific contributions of each isoform to the total cytosolic Hsp90 pool rather than the manifestation of a global stress response.

So far, our results support the conclusion that increased translation of the Hsp90β mRNA accounts for the increased Hsp90β protein levels. To exclude a contribution of reduced turnover of the Hsp90β protein, we performed a cycloheximide-chase assay comparing the turnover of Hsp90β and several Hsp90 client and non-client proteins in 90αKO 90βHET versus 90βHET MAFs. If anything, the results show that the turnover of Hsp90β and the two relatively short-lived Hsp90 clients AKT and c-Raf is higher in 90αKO 90βHET MAFs (Supplementary Fig. 6g). We conclude that an Hsp90β-specific translational activation underlies the increase in Hsp90β protein levels in cells and tissue of the 90αKO 90βHET genotype.

## Increased translation of Hsp90β mRNA is IRES-driven

The reduced global translation rate in 90αKO 90βHET MAFs and brain correlates with the remarkably reduced activity of mTORC1 and hyperphosphorylation of eIF2α (Supplementary Fig. 7a). Since these major mediators of cap-dependent translation[43] are inhibited in 90αKO 90βHET cells, we examined the possibility of cap-independent translation of the Hsp90β mRNA through an IRES. To this end, we used bicistronic expression plasmids where Renilla luciferase is translated in a cap-dependent manner, and firefly luciferase via an IRES from a single mRNA transcript. The 5′-UTR of mouse *Hsp90ab1* was tested for its IRES function in parallel with a well-studied poliovirus IRES[44] as a positive control (Supplementary Fig. 7b). We found not only that the *Hsp90ab1* 5′-UTR does have IRES function, but that this function is increased in 90αKO 90βHET MAFs indicating that the IRES-mediated translational reprogramming may account for the rectification of Hsp90β protein levels (Fig. 3e). We used quantitative RT-PCR to confirm that the IRES-driven enhanced translation of the reporter mRNA in 90αKO 90βHET MAFs is not due to increased levels of the bicistronic reporter mRNA (Supplementary Fig. 7c, d).

To evaluate further the impact of the UTRs of the *Hsp90ab1* mRNA on translation, reporter plasmids were generated where the firefly luciferase coding sequence was flanked either by the 5′- or 3′- or both UTRs (Supplementary Fig. 7e). Remarkably, in 90αKO 90βHET MAFs, the 5′-UTR of the *Hsp90ab1* mRNA by itself enhances firefly luciferase expression (Fig. 3f), again independently of any impact on the steady-state abundance of the corresponding mRNA (Supplementary Fig. 7f). For all other genotypes, both UTRs are required for maximal stimulation of luciferase expression (Fig. 3f). Even though the mechanism of this genotype-dependent interplay remains to be elucidated, this complementary experiment supports the conclusion that

the IRES of the 5′-UTR of the *Hsp90ab1* mRNA potentiates translation more robustly in 90αKO 90βHET cells.

So far, we had investigated the IRES-driven translational reprogramming in cells of the 90αKO 90βHET genotype under normal physiological conditions. We now turned to explore the effects of heat shock, a physiologically relevant stress that inhibits cap-dependent translation[45]. Short-term mild heat stress (30 h at 40 °C) significantly increases the IRES function of the 5′-UTR of the *Hsp90ab1* mRNA only in 90αKO 90βHET and 90βHET MAFs (Supplementary Fig. 7g). A long-term mild heat stress (6 days at 40 °C) remarkably boosts the IRES function even more in 90αKO 90βHET MAFs. During adaptation to long-term heat stress, the 5′-UTR also gained some IRES function in WT MAFs (Supplementary Fig. 7g). To reveal the potential impact of the IRES function of the 5′-UTR of the *Hsp90ab1* mRNA on its endogenous protein production, WT and Hsp90 mutant MAFs were exposed to mild heat stress. Although Hsp90α and Hsp90β mRNA are induced, short-term mild heat stress reduces the steady-state Hsp90 protein levels of both isoforms except in 90αKO 90βHET MAFs (Supplementary Fig. 7h, i). Remarkably, in 90αKO 90βHET MAFs, the fold-induction of Hsp90β protein under short- and long-term heat stress overrides the fold-change of its mRNA (Fig. 3g, h and Supplementary Fig. 7h, i). During the long-term heat adaptation, whereas the protein levels of the Hsp90 isoforms are induced in the survivors with other genotypes, they cannot surpass the fold-change of the levels of the respective mRNA (Fig. 3g, h). Therefore, it is conceivable that the IRES function of the *Hsp90ab1* 5′-UTR not only rectifies the Hsp90β protein levels associated with the 90αKO 90βHET genotype under physiological conditions, but it may also help to fine-tune Hsp90β protein levels in response to stress.

We next sought to determine whether IRES-mediated induction of Hsp90β synthesis in response to stress is conserved in human cells. We tested this with human RPE1 cells, which are a non-cancerous epithelial cell line, upon short-term (30 h) and long-term (4 days) adaptation to mild heat stress. While the stress-induced modest increase in *HSP90AB1* mRNA is remarkably proportional to the corresponding increase in Hsp90β protein, there is a much stronger contribution of the mRNA induction for Hsp90α (Supplementary Fig. 7j). If one considers the fact that general cap-dependent translation is reduced in response to heat stress, it can be hypothesized that the increase in Hsp90β mRNA cannot fully account for the induction of Hsp90β protein, and that there must therefore be a contribution from IRES-dependent translation. The existence of an IRES in human Hsp90β mRNA is supported by the result of a genome-wide screen for IRES[46], and our own results obtained with the bicistronic reporter gene in RPE1 cells, which show a robust IRES activity upon prolonged (5 days) mild heat stress (Supplementary Fig. 7k). Taken together, our data suggest that the increase in both Hsp90α and Hsp90β proteins during short-term adaptation to mild heat stress is primarily transcriptional in normal mammalian cells, whereas the IRES of Hsp90β mRNA becomes important for the adaptation to prolonged heat stress (for MAFs, see also Supplementary Fig. 7g).

## Threshold levels of total Hsp90 for mammalian life

To investigate whether and how the rectification of Hsp90 levels is physiologically relevant, we analyzed the Hsp90β protein levels in postnatal day 1 (P1) pups. Despite the elevated embryonic lethality of the 90αKO 90βHET genotype, we managed to identify two, albeit developmentally retarded, pups to compare with live P1 90αKO pups (Fig. 4a; see also Fig. 1c). We found that the abundance of Hsp90β is remarkably reduced in the stillborn 90αKO 90βHET pups (Fig. 4b). However, the comparison of Hsp90β levels in tissues of adult survivors with the identical genotypes revealed a striking rectification of Hsp90β protein levels in these 90αKO 90βHET mice (Figs. 2a–d and 4b, and Supplementary Figs. 4a and 5a). Therefore, we speculated that below a

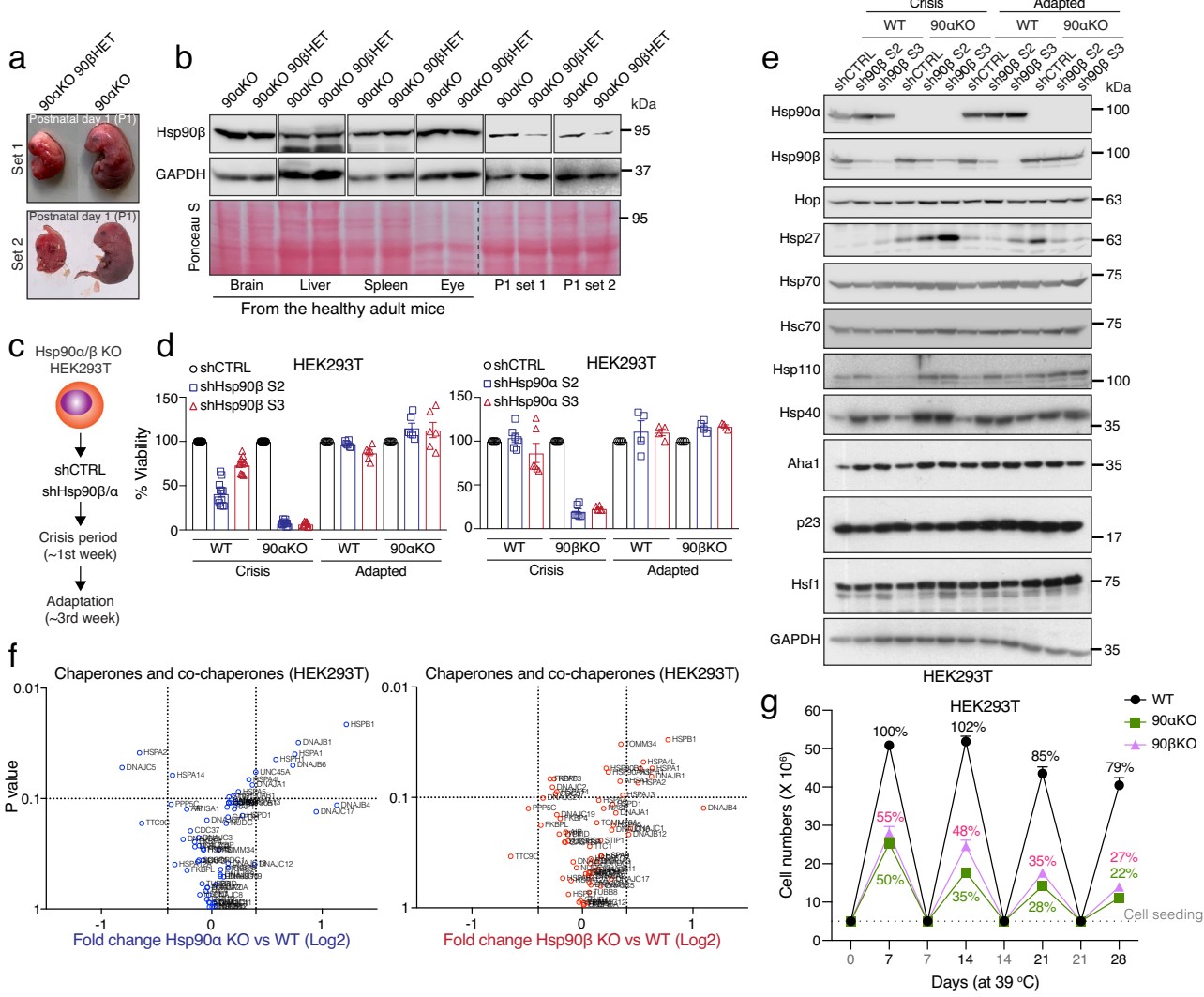

**Fig. 4 | Below threshold levels of Hsp90 are detrimental for mammalian life.**
**a** Representative image of newborn pups of the indicated genotypes at postnatal day 1. 90αKO 90βHET pups are stillborn and developmentally retarded (see also Fig. 1a, c). **b** Immunoblots of Hsp90β from the indicated adult tissues (representative of $n = 4$ biologically independent experiments) or P1 pups (representative of $n = 2$ biologically independent experiments) of the 90αKO and 90αKO 90βHET genotypes; in this experiment, 90αKO P1 pups were live at birth and 90αKO 90βHET P1 pups were stillborn. GAPDH and Ponceau S-stained nitrocellulose membranes serve as loading controls. **c** Strategy for the knockdown (KD) of the remaining Hsp90 isoform in Hsp90α/β knockout (KO) HEK293T cells. **d** Measurement of % cell viability upon KD of the remaining Hsp90 isoform in Hsp90α/β KO HEK293T cells during the crisis period ($n = 12$ and $n = 6$ biologically independent samples for the Hsp90β and Hsp90α KD sets, respectively) and after adaptation ($n = 6$ and $n = 4$ biologically independent samples for the Hsp90β and Hsp90α KD sets, respectively) compared to that of WT (see panel c for the strategy). Two shRNA (S2 or S3) targeting sequences were designed for each Hsp90 isoform. Cell viability of control (shCTRL) sets was set to 100%. **e** Immunoblots of

molecular chaperones, co-chaperones, and stress-related proteins upon KD of Hsp90β in WT and Hsp90α KO HEK293T cells during the crisis period and after adaptation (representative of $n = 2$ biologically independent experiments). GAPDH serves as loading control. **f** Volcano plots of the normalized fold changes of the Hsp70-Hsp90-related chaperones, co-chaperones, and other stress-responsive proteins determined by quantitative label-free proteomic analysis of Hsp90α/β KO and WT HEK293T cells ($n = 3$ biologically independent samples). Log2 fold change of >0.4 or <−0.4 with a p-value of <0.1 were considered significant differences for a particular protein. Comparisons between samples of different genotypes are shown in different colors. p-values were calculated by a two-tailed unpaired Student's t-test with Benjamini-Hochberg p-value correction. **g** Cell proliferation of WT and Hsp90α/β KO HEK293T cells at 39 °C for the indicated time presented as cell numbers ($n = 5$ biologically independent experiments). Cells were reseeded at the density of $5 \times 10^6$ every 7th day. The number of WT cells after the first 7 days in culture was set to 100%. The data of panels d and g are represented as mean values ± SEM for the bar and line graphs, respectively. Source data are provided as a Source Data file.

certain threshold level, Hsp90 may not support the viability of a mammalian organism and that the rectified Hsp90 levels define a criterion for mammalian life.

To further explore this hypothesis in a more controllable cellular model, we knocked down the remaining isoforms of Hsp90 in Hsp90α (encoded by the gene *HSP90AA1*) and Hsp90β (encoded by *HSP90AB1*) knockout (KO) HEK293T cells[38] (Fig. 4c, Supplementary Data 2). Further reduction of Hsp90 levels in Hsp90α/β KO HEK293T cells led to a remarkable lethality during the "crisis" period, that is the initial phase

of the knockdown (Fig. 4d, and Supplementary Fig. 8a, b). Consistent with our genetic models, functional inhibition of Hsp90 by geldanamycin (GA) revealed markedly enhanced cytotoxicity for all Hsp90 mutant cell lines from both mouse and human origins (Supplementary Fig. 8c–f).

After the initial knockdown of the remaining Hsp90 isoform and the associated "crisis" period, a small number of Hsp90α/β KO HEK293T cells were clonally selected and started growing normally (Fig. 4d, and Supplementary Fig. 8a, b). Investigating the molecular

basis of the long-term adaptation of Hsp90α/β KO HEK293T cells, we found that the initially reduced levels of the targeted Hsp90 isoform were largely restored both at the protein and the mRNA levels, whereas the knockdown remained stable in WT cells (Fig. 4e, and Supplementary Fig. 8g, h). These data, obtained with a totally orthogonal experimental system, support our conclusion that the viability of a mammalian cell or organism can only be supported by above-threshold total Hsp90 levels, irrespective of the isoform.

We performed a proteomic analysis to quantitate the Hsp90 threshold levels in these Hsp90 KO lines more accurately. This revealed that in adapted cells, compared to the total Hsp90 levels of WT cells, the remaining Hsp90 isoform reaches levels of about 75% and 50% in Hsp90α and Hsp90β KO HEK293T cells, respectively (Supplementary Fig. 8i, Supplementary Data 2). Reducing these levels further causes severe lethality in Hsp90α/β KO HEK293T cells, as described above (Supplementary Fig. 8a, b, j and Fig. 4d). Therefore, at the cellular level of this mammalian model system, 50–75% of the total Hsp90 is required for viability. Quantitative proteomic analysis of tissues from WT and Hsp90 mutant mice showed that the minimum levels of total Hsp90 might vary depending on tissue-specific requirements (Fig. 2b). However, none of the analyzed tissues, notably of the 90αKO 90βHET genotype, showed <50% of total Hsp90 (Fig. 2b), far above the 1–5% required to support yeast viability[33,34]. The incompressible part of the steady-state levels of Hsp90 may have increased to accommodate the ever-growing complexity of the proteome[22] on the path towards mammals.

## Other chaperones are unable to rescue the lethality of the Hsp90 deficiency

Hsp90 inhibitors induce the stress response and as a consequence the overexpression of other molecular chaperones and co-chaperones. This is a critical mechanism for adaptation and acquisition of drug resistance of cancer cells[47]. Therefore, we wondered whether other stress-related proteins contribute to the adaptation of Hsp90α/β KO HEK293T cells after the initial knockdown of the remaining Hsp90 isoform. During the long-term adaptation, the levels of stress-related proteins, which are initially increased in Hsp90α/β KO HEK293T cells, revert back to their respective basal levels, in parallel with the rectification of total Hsp90 levels (Fig. 4e, and Supplementary Fig. 8g). To determine whether the adapted state is stable, we subjected the same cells to a second round of Hsp90β knockdown. Again, while this caused a severe lethality within the first week, during adaptation, a few survivors managed to increase the initially reduced Hsp90β levels (Supplementary Fig. 9a–c).

While analyzing the levels of Hsp90 in the aforementioned experiments with immunoblots, we noticed that the basal levels of several stress-related proteins, including Hsp70, Hsp40, Hsp27, and Aha1, are elevated in Hsp90 KO cells (Fig. 4e, Supplementary Fig. 8g). This could be confirmed for an additional cell line (A549), in which we reduced Hsp90 levels with the CRISPR-Cas9 system (Supplementary Figs. 8e and 9d), and by quantitative proteomic analyses of the HEK293T KOs (Fig. 4f, Supplementary Data 2) and of liver and muscle of 90αKO 90βHET mice (Supplementary Fig. 9e, f, Supplementary Data 1). In contrast, clients and other interactors of Hsp90 do not significantly change in any of the Hsp90 mutant cells and tissues compared to WT under normal physiological conditions (Supplementary Fig. 9g, h, Supplementary Data 1 and 2). To evaluate further whether the above-mentioned increased basal level of some stress-related proteins can compensate for the reduced Hsp90 levels, WT and Hsp90 KO/mutant HEK293T and A549 cells were exposed to a long-term mild heat stress. All cell lines with reduced Hsp90 levels proved to be much more sensitive than the WT parent cell line (Fig. 4g, and Supplementary Fig. 10a–d). Taken together, all of these results demonstrate that the increased expression of other molecular chaperones cannot compensate for the lack of Hsp90 under stress

conditions and that they cannot substantially contribute to the adaptation of Hsp90-deficient cells.

While the results of the different types of loss-of-function experiments were consistent across the board, we performed a rescue experiment to provide further support for our conclusions. Transient overexpression of either one of the Hsp90 isoforms in Hsp90α/β KO HEK293T cells significantly improved cell viability under mild heat stress (Supplementary Fig. 10e–g). These results indicate that environmental conditions determine the Hsp90 requirements for mammalian life, and that the total abundance of cytosolic Hsp90 is the important parameter.

## Below threshold levels of Hsp90 in mammals cause a proteotoxic collapse

We next investigated the molecular mechanism behind the premature death of 90αKO 90βHET embryos during gestation. Although the basal levels of Hsp90β are similar in survivors with the 90αKO and 90αKO 90βHET genotypes (Fig. 2c, d and Supplementary Figs. 4a and 5a), short-term pharmacological inhibition of the remaining Hsp90 leads to an increased accumulation of total and polyubiquitinated insoluble proteins in 90αKO 90βHET MAFs (Fig. 5a). Reminiscent of these findings, stillborn 90αKO 90βHET pups also accumulate more total and polyubiquitinated insoluble proteins than live 90αKO P1 pups (Fig. 5b, and Supplementary Fig. 11a). As mentioned before, stillborn 90αKO 90βHET pups have low Hsp90β levels compared to both 90αKO P1 pups and 90αKO 90βHET adult survivors (Fig. 4a, b). The latter did not show any notable differential accumulation of total and polyubiquitinated insoluble proteins in the liver by comparison to 90αKO mice (Fig. 5b and Supplementary Fig. 11a). The accumulation of insoluble proteins can also be observed in Hsp90α/β KO HEK293T cells subjected to a long-term mild heat stress. When these cells are returned to normal temperature, they recover their ability to clear these protein aggregates (Fig. 5c). Moreover, increased levels of total Hsp90 by transient overexpression of either one of the Hsp90 isoforms in these Hsp90 KO cells significantly reduced proteotoxicity as evidenced by reduced heat stress-induced accumulation of total and polyubiquitinated insoluble proteins (Supplementary Fig. 11b). This finding parallels the improved cell viability of Hsp90α/β KO HEK293T cells under similar experimental condition (see Supplementary Fig. 10g).

Hsp90 has been proposed to be a capacitor of morphological evolution by buffering preexisting genetic polymorphisms[48–51]. The ability of Hsp90 to restrain the expression of transposable elements, including cellular retroviral genes, is thought to contribute to both short- and long-term buffering against phenotypic changes[52]. Since we did not find any upregulation of two such cellular retroviral genes (MERVL and IAPEz)[52] in the stillborn 90αKO 90βHET pups, it appears that the capacitor function of Hsp90 remained intact in these mutant mouse models (Supplementary Fig. 11c). Therefore, we conclude that below threshold levels of total cytosolic Hsp90 lead to the death of mouse embryos and mammalian cells via proteotoxicity.

## Maintaining the levels of Hsp90 may delay aging by attenuating the expression of senescence markers

Mammalian aging correlates with proteostatic collapse, including compromised molecular chaperone levels and functions[16,17,53–55]. Since cellular senescence is a crucial biological process underlying aging[56], we investigated whether below threshold levels of Hsp90 trigger accelerated senescence. Knocking down the remaining Hsp90 isoform in Hsp90 α/β KO HEK293T cells (see Fig. 4c for strategy) increased several senescence markers, including p21, p16, and p27 (encoded by the CDKN1A, CDKN2A, and CDKN1B genes, respectively) during the crisis period (Fig. 5d, and Supplementary Fig. 11d). However, during long-term adaptation these senescence markers were reduced to the basal levels in parallel with the re-expression of the initially knocked-down

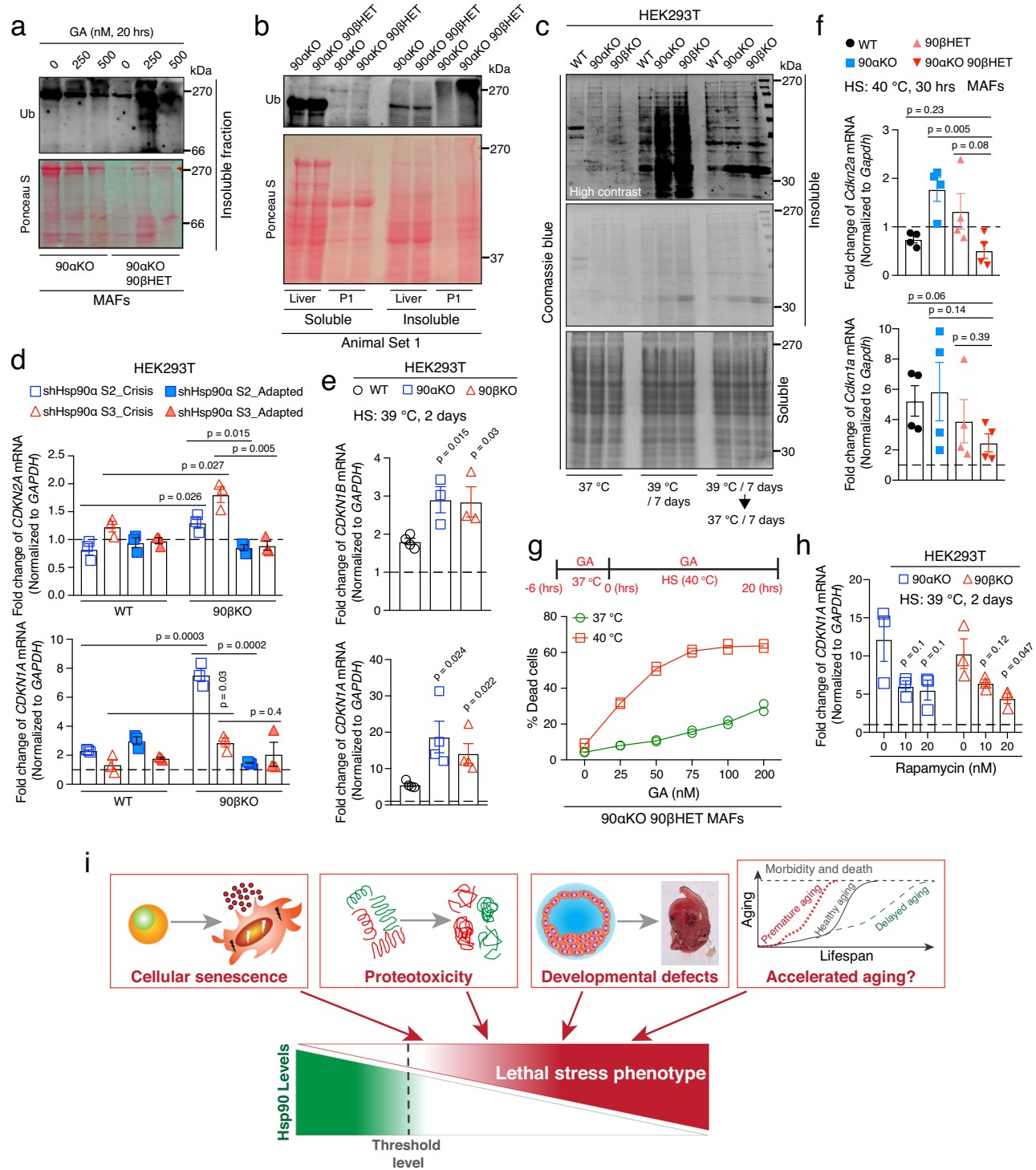

Hsp90 isoform (Fig. 5d and Supplementary Fig. 11e). Hence, below threshold levels of Hsp90 appear to accelerate senescence, and they may therefore accelerate mammalian organismal aging, too.

Using mild heat stress as a stimulus to accelerate senescence, we found that Hsp90α/β KO HEK293T cells are more prone to becoming senescent than WT cells (Fig. 5e). These results further support our earlier findings of reduced growth and enhanced death of Hsp90α/β KO HEK293T cells under similar stress conditions (Fig. 4g and Supplementary Fig. 10b). Therefore, maintaining Hsp90 above threshold levels attenuates senescence and may delay accelerated aging.

Next, we investigated whether mammalian cells can actively tune Hsp90 levels, thereby delaying aging. We found that senescence

induced by a short-term mild heat stress is remarkably low in 90αKO 90βHET MAFs (Fig. 5f and Supplementary Fig. 12a), which overexpress the remaining isoform Hsp90β from the beginning (see Supplementary Fig. 7h, i). When subjected to a long-term mild heat stress, these MAFs grew best and displayed the most normal morphology (Supplementary Fig. 12b, c). Prior pharmacological inhibition of Hsp90 with GA suppresses the superior Hsp90-dependent cellular fitness of 90αKO 90βHET MAFs under mild heat stress, which leads to increased cell death when combined with continued exposure to GA (Fig. 5g and Supplementary Fig. 12d). We conclude from this series of experiments that mammalian organisms and cells that cannot fine-tune Hsp90 to the physiologically required levels might experience accelerated aging

**Fig. 5 | Below threshold levels of Hsp90 augment proteotoxic collapse and cellular senescence. a** Aggregation of total (Ponceau S-stained proteins) and polyubiquitinated proteins in geldanamycin (GA)-treated 90αKO and 90αKO 90βHET MAFs (representative of $n = 2$ biologically independent experiments). Note that the apparent reduction of polyubiquitinated materials at 500 nM GA compared to 250 nM may be due to GA-induced enhanced degradation of E3-ligases themselves, since many of them are Hsp90 clients[92]. **b** Total (Ponceau S-stained proteins) and polyubiquitinated detergent-soluble and -insoluble proteins from the liver of adult mice and P1 pups of the indicated genotypes (representative of $n = 2$ biologically independent experiments). This particular 90αKO 90βHET P1 pup was stillborn (see Fig. 4a). **c** Analysis by Coomassie blue staining of an SDS-PAGE of HS (39 °C for 7 days)-induced accumulation of detergent-insoluble aggregated proteins and subsequent clearance upon returning to 37 °C in WT and Hsp90α/β KO HEK293T cells (representative of $n = 2$ biologically independent experiments). **d** mRNA expression of the senescence markers *CDKN2A* (p16) and *CDKN1A* (p21) upon KD of Hsp90α in Hsp90β KO HEK293T cells during the crisis period and after adaptation compared to that of WT (see Fig. 4c for the strategy) ($n = 3$ biologically

independent samples). shCTRL sets were set to 1 (dashed line). **e** HS (39 °C for 2 days)-induced mRNA expression of the senescence markers *CDKN1B* (p27; WT, $n = 4$; Hsp90α/β KO, $n = 3$ biologically independent samples) and *CDKN1A* (p21; $n = 4$ biologically independent samples for all genotypes) in WT and Hsp90α/β KO HEK293T cells. **f** HS (40 °C for 30 h)-induced mRNA expression of *Cdkn2a* (p16) and *Cdkn1a* (p21), in MAFs of the indicated genotypes ($n = 4$ biologically independent samples). **g** Flow cytometric quantification of cell death of 90αKO 90βHET MAFs at 40 °C and 37 °C in the absence and presence of GA ($n = 2$ biologically independent samples). **h**, Impact of rapamycin on the HS (39 °C for 2 days)-induced mRNA expression of *CDKN1A* (p21) in Hsp90α/β KO HEK293T cells ($n = 3$ biologically independent samples). *GAPDH/Gapdh* was used as a reference gene in all mRNA expression analyses. mRNA expression at 37 °C was set to 1 (dashed line) in panels **e**, **f**, and **h**. The data are represented as mean values ± SEM for the bar and line graphs. The statistical significance between the groups was analyzed by two-tailed unpaired Student's *t*-tests. **i** Schematic representation of the physiological relevance of incompressible Hsp90 levels for mammalian life. Source data are provided as a Source Data file.

and premature death, such as those 90αKO 90βHET embryos, which fail to increase Hsp90β levels. Several molecular chaperones, including Hsp90, are known to be reduced during aging in mammals, including in humans[53–55]. We found that the mTOR inhibitor and anti-aging drug rapamycin[57,58] reduces the heat stress-induced senescence marker p21 in Hsp90α/β KO HEK293T cells (Fig. 5h), consistent with the idea that it may be therapeutically beneficial to partially inhibit the growth stimulatory mTOR signal to overcome premature aging.

## Discussion
Survival is the fundamental criterion of living organisms. To do so, organisms must be able to adapt to intrinsic and environmentally imposed stress by regulating several biological processes, collectively referred to as the proteostasis system[3,17,59]. Molecular chaperones, including the Hsp90 chaperone system, are both the sensors and effectors of stress responses[3,17,60]. Organisms that fail to respond to stress are eliminated, and therefore, stress-response mechanisms affect natural selection and evolution[1,9,51]. Although the reduced levels of molecular chaperones, such as Hsp90, likely have an impact on the evolution of species, whether organisms can actively act on the basal levels of Hsp90 to ensure survival and reproduction was unclear.

Here we report, using different Hsp90 mutant mouse and cellular models, that eukaryotic organisms can actively manipulate Hsp90 levels to support development and survival (Fig. 5i). Organisms that are unable to maintain Hsp90 above cell- and tissue-specific threshold levels may not be naturally selected. Our data support the conclusion that declining proteostasis and accelerated senescence are the primary causes of death in organisms with reduced levels of Hsp90 (Fig. 5i). The fact that the expression of two mammalian retroviral genes (MERVL and IAPEz) was not unleashed in 90αKO 90βHET embryos suggests that genetic variations generated de novo by transposable elements may not account for the embryonic lethality associated with this genotype. However, more in-depth analyses of the dynamics of transposable elements and of other genomic alterations might help to clarify whether they need to be considered or whether the progressive deterioration of proteostasis is sufficient by itself to explain embryonic lethality.

Hsp90 has been said to be the most abundant soluble protein of eukaryotic cells, contributing 1–2% of the total cellular proteome[20,61]. We speculate that there are two different pools in this enormous amount of Hsp90 protein, a functionally "active" and a "latent" pool. The absolute and relative amounts of these pools may be specifically set for each tissue and cell type. We assume that the "active" pools determine the threshold levels for survival. Since these basal levels can be assumed to be "constitutively active," any chemical or genetic perturbation of these levels would be highly detrimental. Our data clearly demonstrate that these threshold levels are significantly higher

in mammalian cells than in lower eukaryotes such as in the budding yeast[34]. This supports our initial hypothesis that the evolution to more complex proteomes imposed a need for considerably higher minimal levels of Hsp90 and possibly other molecular chaperones.

The transcriptional upregulation of the mRNA transcripts for molecular chaperones is a well-known phenomenon amongst stress response mechanisms, including for the cytosolic heat shock response[60] and the endoplasmic reticulum-specific unfolded protein response[62]. Diverse extrinsic or intrinsic stress stimuli elicit a eukaryote-specific adaptive response, which has been termed the integrated stress response (ISR)[12,63,64]. In the ISR, global cap-dependent protein translation is specifically reduced by the hyperphosphorylation of eIF2α[63]. However, a subset of cellular proteins necessary for the ISR escape this inhibition through cap-independent mechanisms, including by IRES-driven translation[65,66]. IRES, which were first discovered in viral mRNAs, were also found in several cellular genes and are responsible for selective protein expression when cap-dependent translation is inhibited, including during mitosis, apoptosis, cell differentiation, and angiogenesis[65–68]. We found that the 5′-UTR of the mouse *Hsp90ab1* mRNA possesses IRES function and that this function is specifically required for the rectification of Hsp90β protein level in cells and tissues with the 90αKO 90βHET genotype where the global cap-dependent translation is markedly reduced. It is conceivable that genetic, metabolic or pharmacological perturbations of Hsp90 levels and functions may trigger aspects of the ISR.

Cellular IRES functions are strongly influenced by IRES-transacting factors (ITAFs)[65,66,68]. We speculate that some ITAFs may differentially affect the IRES function of the 5′-UTR of the mouse *Hsp90ab1* mRNA, notably in the context of the 90αKO 90βHET genotype and under stressed conditions. In addition, post-transcriptional modifications of the 5′-UTR, for example, by $N^6$-methyladenosine (m6A) mRNA methylation[69], may also impact the cap-independent translation. This mechanism has recently been revealed for heat-stressed conditions using the molecular chaperone Hsp70 and Dnajb4 as models[70,71]. It is conceivable that specific m6A "writers", "readers", or "erasers"[72] may positively influence the IRES-driven expression of Hsp90β in the 90αKO 90βHET genotype. The identification of these additional molecular players, some of which could themselves be Hsp90 clients, will be necessary to explain why only very few 90αKO 90βHET mouse embryos manage to translationally rectify their Hsp90β protein levels sufficiently.

As mentioned above, because the 5′-UTR of the human *HSP90AB1* mRNA also carries an IRES, it can be expected that the inhibition of global cap-dependent translation may not have a strong negative impact on the expression of human Hsp90β. If this could be confirmed, temporary inhibition of global cap-dependent translation by therapeutic drugs such as rapamycin or other rapalogs might be

beneficial to treat aging and other protein-misfolding diseases in humans. Rapamycin acts by increasing autophagy and decreasing cellular senescence and senescence-associated secretory phenotypes[57,73–75]. It is noteworthy that all axes of proteostasis, including molecular chaperones, are overwhelmed by the accumulation of "difficult-to-fold" or "difficult-to-degrade" proteins and aggregates in all degenerative physiological states, including during aging[16,17,76]. We speculate that rapamycin treatment may elevate the ratio of Hsp90β to substrate, and possibly that of Hsp90α to substrate if its mRNA can be shown to display some IRES activity. This would help overcome the proteostatic decline associated with aging and other protein misfolding disorders.

Aging is the gradual degeneration of physiological states responsible for age-related morbidity and mortality[77]. Aging is collectively the consequence of proteostatic collapse, cellular senescence, genomic instability, epigenetic reprogramming, inflammatory responses, and many more[16,17,77]. Here we have connected aging to reduced levels of Hsp90, which cause proteotoxicity and accelerated cellular senescence. In apparent contradiction, Hsp90 inhibitors have been proposed as senolytic drugs[78], which can selectively kill senescent cells, thereby rejuvenating the organism[79]. Although below-threshold levels of Hsp90 drive accelerated cellular senescence, we speculate that this residual and inhibitable Hsp90 still supports the senescence-associated phenotypes. It would be fascinating to find out in the future whether this "alternate" Hsp90 function is directed by post-translational modifications of Hsp90 or differential co-chaperone influence or the formation of an alternative Hsp90 interactome, an Hsp90 "epichaperome" as suggested for cancer and neurodegenerative diseases[80,81].

# Methods

## Ethics statement
Animal breeding and all animal experiments were carried out according to Swiss laws and with formal authorizations (animal experimentation licences GE/55/15, GE/179/18, and GE/180/19) from the State (Direction générale de la santé, République et Canton de Genève) and Federal (Office fédéral de la sécurité alimentaire et des affaires vétérinaires) authorities.

## Mouse lines, breeding, and genotyping
Our Hsp90β mutant mice derive from embryonic stem (ES) cells of the C57BL/6 N strain background (subclone JM8.N4) with a targeted mutation of one allele of the mouse *Hsp90ab1* gene (mutant allele Hsp90ab1tm1a(EUCOMM)Wtsi; for details, see "Mouse Genome Informatics" entry http://www.informatics.jax.org/allele/MGI:4433179, and Supplementary Fig. 1a). We obtained several of these mutated ES cell clones of project 31882 of the European Conditional Mouse Mutagenesis Program (EUCOMM) located at the Helmholtz-Zentrum München. Chimeras were obtained from two ES cell clones by blastocyst injection performed by Polygene (https://www.polygene.ch). Chimeric males were bred to WT C57BL/6N females to obtain germline transmission. Heterozygous mice were crossed with a ROSA26::FLPe deleter strain (a gift from Dmitri Firsov's group at the University of Lausanne; originally from Cyagen and in the C57BL/6N strain background), using both male and female mice of 3–8 months of age, to remove the FRT-flanked gene trap construct with a βGEO cassette from the targeted *Hsp90ab1* allele (Supplementary Fig. 1a). Subsequently, the floxed exons 2–6 of the *Hsp90ab1* gene were deleted by crossing to mice with a transgene for Cre recombinase expression under the control of the CMV enhancer/promoter (a gift from the group of Ivan Rodriguez at the University of Geneva), thereby generating heterozygous *Hsp90ab1* mutant mice (90βHET) (Supplementary Fig. 1a).

*Hsp90aa1* knockout (90αKO) mice had previously been established and characterized in our laboratory[39]. In order to get different KO/HET combinations of Hsp90 alleles, we performed different breeding experiments as illustrated in Supplementary Fig. 1c–g. Since 90αKO male mice are sterile[39], we could only use 90αKO female mice for the relevant breeding experiments. PCR analyses confirmed mouse genotypes with DNA isolated from ear biopsies using the KAPA Mouse Genotyping Kits (Kapa Biosystems). All PCR primers are listed in Supplementary Table 1.

Mice were housed in a licensed animal house at 21–23 °C, 45–55% humidity, and with a 12 h/12 h dark/light cycle.

## Cell lines and cell culture
HEK293T human embryonic kidney cells (ATCC, CRL-3216), A549 human lung epithelial carcinoma cells (ATCC, CCL-185) (as well as the corresponding Hsp90α/β KO or mutant cell lines), and RPE1 human retinal epithelial cells (ATCC, CRL-4000) were maintained in Dulbecco's Modified Eagle Media (DMEM) supplemented with GlutaMAX, 10% fetal bovine serum (FBS), and penicillin/streptomycin (100 u/ml) with 5% $CO_2$ in a 37 °C humidified incubator.

To establish the culture of MAFs, ear biopsies of WT and Hsp90 mutant adult mice (both male and female animals of 7–9 months of age) were cut into small pieces using scalpels and incubated overnight in a digestion buffer (Roswell Park Memorial Institute (RPMI) medium, 1 mg/ml collagenase, 30% FBS, 1% L-glutamine, penicillin/streptomycin (100 u/ml)) in a 37 °C humidified incubator. Single-cell suspensions were seeded and cultured in RPMI medium supplemented with 20% FBS, 1% L-glutamine, and penicillin/streptomycin (100 u/ml). After several passages, when cells started growing normally and were not dying anymore, cells were considered to be immortalized MAFs and switched to DMEM supplemented with GlutaMAX, 20% FBS, and penicillin/streptomycin (100 u/ml). Experiments related to MAFs were performed with the cells at 12–24 passages.

## Plasmids
Bicistronic reporter plasmid pcDNA3 RLUC POLIRES FLUC (a gift from Nahum Sonenberg)[82] was obtained from Addgene (#45642). To evaluate IRES functions of the 5´-UTR of mouse *Hsp90ab1*, 105 nucleotides immediately upstream of the translation start codon (transcript ID ENSMUST00000024739.14) was PCR-amplified using cDNA from 90αKO 90βHET mouse brain as a template and cloned between HindIII and BamHI sites of plasmid pcDNA3 RLUC POLIRES FLUC such that the Polio IRES sequence was excised. The 5′-UTR of human *HSP90AB1* (214 nucleotides immediately upstream of the start codon of transcript ENST00000371554.2) was PCR-amplified using cDNA from HEK293T cells and similarly cloned into the recipient plasmid. The resultant plasmid is referred to as pcDNA3 RLUC 5′-UTR FLUC. Note that nucleotides 182–184 are TCG instead of AGA and correspond to the sequence present in some polymorphic variants deposited in GenBank. With these reporter plasmids, Renilla luciferase, which serves as an internal control, is expressed by cap-dependent translation and firefly luciferase in a cap-independent manner via IRES from a single mRNA transcript.

To evaluate the impacts of the UTRs of mouse *Hsp90ab1* on the translation of a reporter gene, the firefly luciferase coding sequence was flanked either by the 5′- or 3′- or both UTRs. Briefly, 105 nucleotides immediately upstream of the translation start codon and 240 nucleotides immediately downstream of the stop codon (transcript ID ENSMUST00000024739.14) were PCR-amplified using cDNA from 90αKO 90βHET mouse brain as a template. 5′- and 3′-UTRs nucleotide sequences were inserted into NcoI and XbaI sites, respectively, of plasmid pGL3-CMV.Luc. All inserted UTR sequences were verified by DNA sequencing.

## Genome engineering
Human Hsp90α (*HSP90AA1*) and Hsp90β (*HSP90AB1*) KO/mutant A549 cells were generated by the CRISPR/Cas9 gene-editing technology as reported earlier for Hsp90α/β KO HEK293T cells[38]. Individual

targeted cell foci were picked, expanded, and analyzed by immuno-blotting using primary antibodies specific to Hsp90α and Hsp90β. Clones that did not express or expressed significantly reduced levels of Hsp90α or Hsp90β compared to corresponding WT cells were considered KO/mutant cells. Hsp90α/β KO clones from HEK293T cells were also validated by MS analysis. Hsp90 mutant A549 clones were designated as "sg90α/β(clone number)" since the complete absence of full-length Hsp90 protein was only validated by immunoblotting but not by mass spectrometric analysis.

## Induction of heat shock
To determine the impact of heat shock on cell proliferation and viability, WT and Hsp90α/β KO/mutant HEK293T and A549 cells were seeded at a density of $3-5 \times 10^6$ cells per 20 ml and subjected to mild heat shock at 39 °C and 40 °C for HEK293T and A549 cell clones, respectively, with 5% $CO_2$ in a humidified incubator. A parallel set was maintained at 37 °C as control. Every 7 days, cells were harvested by trypsinization, counted, and reseeded at the same density of initial seeding. This cycle continued for 3–4 weeks from the beginning of the heat shock induction. To evaluate the effect of heat shock on cell death WT and Hsp90α/β KO/mutant HEK293T and A549 cells were seeded at a density of $2.5 \times 10^5$ cells per 3 ml and cultured as described above for 4–7 days. To evaluate heat shock-induced senescence, WT and Hsp90α/β KO HEK293T cells were seeded at a density of $2 \times 10^5$ cells per 3 ml and cultured at 39 °C for 2 days. In some experiments, rapamycin (0–20 nM) was added during the heat shock (for 2 days) to check its impact on heat shock-induced senescence. A parallel set was maintained at 37 °C as control. MAFs were seeded at a density of $5 \times 10^5$ cells per 10 ml and cultured at 40 °C or 37 °C either for 30 h or 7 days. Subsequently, MAFs were harvested for immunoblot or quantitative RT-PCR analyses or cell counting experiments. Heat shock-induced changes were always compared to the time-matched 37 °C control sets.

## Pharmacological inhibition of biological processes
Assays were performed with Hsp90 WT or KO/mutant human cells and MAFs. For immunoblot analyses of soluble and insoluble fractions, MAFs were seeded at a density of $1.5 \times 10^6$ cells per 10 ml and treated with GA (0–500 nM) for 20 h. To determine the impact of Hsp90 inhibition on cell death, human cells or MAFs were seeded at a density of $4 \times 10^5$ per 2 ml or $2 \times 10^5$ per 3 ml, respectively, and treated with GA (0–750 nM) for 48 h. In another experiment, MAFs were seeded at a density of $5 \times 10^5$ per 10 ml and treated with GA (0–200 nM) for 6 h at 37 °C. Subsequently, one set was kept at 40 °C and another set at 37 °C for another 20 h. Cell death and cellular morphology were further evaluated.

## Lentiviral particle generation and gene knockdown
To generate lentiviral particles, HEK293T cells were co-transfected with plasmids pLKO.1shRNA (5 µg), PMDG.2 (1.25 µg), and psPAX.2 (3.75 µg) with PEI (1:3 DNA to PEI ratio). A non-targeting shRNA (not known to target any human gene) expressed from plasmid pLKO.1 was similarly used to generate lentiviral control particles. Suspensions of lentiviral particles were collected and added to the medium of WT and Hsp90α/β KO HEK293T cells to knock down the expression of specific genes. Transduced cells were selected by puromycin (3–4 µg/ml) and used for further experiments. Gene knockdowns were validated by immunoblot and quantitative RT-PCR analyses. shRNA sequences are listed in Supplementary Table 1.

## Flow cytometry
For all flow cytometric analyses, a minimum of 10,000 cells were analyzed for each sample. We used a FACS Gallios flow cytometer (Beckman Coulter), and data were analyzed with the FlowJo software package. The FACS gating and analysis strategies are shown in Supplementary Fig. 13. Additional details are given in the following paragraphs.

Cell death assays: Following a specific treatment, mouse and human cells were harvested by trypsinization, washed in phosphate-buffered saline (PBS), and resuspended in 100–200 µl PBS containing propidium iodide (PI; 2.5 µg/ml) for 15–20 min at room temperature (RT) before flow cytometric analysis.

Cell cycle analyses: Cells were harvested as detailed above. Next, cells were fixed with 70% ice-cold ethanol, washed in PBS, treated with 100 µg/ml RNase A at RT for 5 min, then incubated with 50 µg/ml PI for 15–20 min at RT before flow cytometric analysis. Apoptotic cells were identified by the quantitation of the SubG0 (<2n DNA) cell population.

## Cell viability assay
Cell viability assays were performed with the CellTiter-Glo (CTG) luminescent assay (Promega) according to the manufacturer's instructions. Briefly, cells were seeded at a density of 5000 cells per 200 µl complete medium in 96-well plates. After seeding cells were allowed to grow for the next 72–96 h. 30 µl of CellTiter-Glo reagent was added per well, and the luminescence was measured with a Cytation 3 microplate reader (BioTek). The luminescence from the control cells was set to 100% viability.

## Assay of protein translation rate
WT and mutant MAFs were seeded at a density of $1.2-1.5 \times 10^6$ per 10 ml and were treated with puromycin (1 µM) for 0–2 h. After treatment cells were harvested and lysed in a lysis buffer (20 mM Tris-HCl pH 7.4, 2 mM EDTA, 150 mM NaCl, 1.2% sodium deoxycholate, 1.2% Triton-X100, 200 mM iodoacetamide, protease inhibitor cocktail [PIC]). 75 µg of clarified cell lysates were separated by SDS-PAGE, and immunoblotted for newly synthesized proteins or polypeptides with anti-puromycin antibodies. The translation rate of Hsp90β was analyzed by quantitating the incorporation of puromycin into newly synthesized Hsp90β over time. For this, nascent Hsp90β polypeptide chains were immunoprecipitated and subsequently revealed by immunoblotting with anti-puromycin antibodies. A relatively small proportion of newly synthesized proteins compared to the steady-state levels, and puromycin-induced premature termination of polypeptide elongation explain the apparently weaker bands of puromycin-labeled full-length Hsp90β. Only full-length (or nearly full-length) puromycin-labeled Hsp90β was considered for further analysis. 0 h puromycin treatment served as a negative control of puromycin labeling. The relative rate of Hsp90β translation to global translation was calculated from the densitometric scores. The rate of puromycin incorporation into newly synthesized proteins is directly proportional to the global rate of translation.

## Cycloheximide chase assay
MAFs were seeded at a density of $4 \times 10^5$ cells per 4 ml and treated with cycloheximide (100 µg/ml) for 0–24 h to evaluate protein turnover rate. Cells were further processed for immunoblot analyses.

## Ribosome and polysome fractionation
For ribosome fractionation, MAFs were lysed in a lysis buffer (10 mM HEPES pH 7.4, 100 mM KCl, 5 mM $MgCl_2$, 100 µg/ml cycloheximide, 1 mM DTT, 2% Triton X-100, PIC). 1.5–2 mg of total protein were loaded on a 60% sucrose cushion (prepared in the same lysis buffer) and fractionated by ultracentrifugation at 70,000 $\times g$ for 3 h at 4 °C. Ribosome precipitates were washed with lysis buffer and processed for further experiments.

Polysome profiling of murine brain tissues (from both male and female animals of 4–5 months of age) was performed as described before[83,84] with some modifications. In brief, snap-frozen 100–150 mg brain tissue were pulverized under liquid nitrogen. Tissue powders were resuspended in a lysis buffer (50 mM Tris-HCl pH 7.4, 100 mM

KCl, 1.5 mM $MgCl_2$, 1 mM DTT, 1 mg/ml heparin, 1% Triton X-100, 0.5% sodium deoxycholate, 100 µg/ml cycloheximide, PIC, 100 U/ml SuperaseIn RNase inhibitor) for 15 min on ice. RNA amounts were quantified in the clarified tissue extracts, and 180 µg total RNA equivalents were loaded onto 20–60% linear sucrose density gradients (prepared in 40 mM HEPES pH 7.5, 40 mM KCl, 20 mM $MgCl_2$). Fractionation was performed by ultracentrifugation (Optima XPN-100, Beckman; SW-41Ti swinging bucket rotor) at $210,100 \times g$ for 3.5 h at 4 °C. The fractions ($12 \times 1$ ml for each sample) were collected with a Foxy R1 fraction collector (ISCO), coupled with a UA-6 absorbance detector equipped with chart recorder (ISCO), and the profiles were recorded with the TracerDAQ Pro data acquisition software (MCC). The polysome profiles derived from the liver samples served as a standard to correctly trace the fraction positions of the "less prominent" brain polysome profiles. To extract polysome-associated RNA, 500 µl of each polysomal fraction (numbers 7–12) were precipitated with 3 volumes of ice-cold 100% ethanol overnight at −80 °C. After centrifugation the pellets from identical samples were pooled and processed for RNA extraction using the acid guanidinium thiocyanate-phenol-chloroform method.

## Mouse tissue extract preparation

Tissues were collected from euthanized mice (both male and female animals of 3–5 months of age), cut into multiple pieces, snap-frozen in liquid nitrogen, and stored at −80 °C. To prepare whole tissue extracts, frozen tissues were thawed on ice and resuspended in a lysis buffer (20 mM Tris-HCl pH 7.4, 2 mM EDTA, 150 mM NaCl, 1.2% sodium deoxycholate, 1.2% Triton-X100, PIC), and homogenized with a MEDIC●TOOL apparatus (Axonlab). Tissue homogenates were sonicated (high power, 20 cycles of 30 s pulses) and centrifuged at $16,100 \times g$ for 20 min at 4 °C. Clarified tissue extracts were used for immunoblotting analyses.

## Biochemical fractionation of soluble and insoluble proteins

After completion of the specific treatments, cells were lysed in a lysis buffer with mild detergents (20 mM Tris-HCl pH 7.4, 2 mM EDTA, 150 mM NaCl, 1.2% sodium deoxycholate, 1.2% Triton-X100, 200 mM iodoacetamide, PIC), sonicated (low power, 6–10 cycles of 20 s pulses), and centrifuged at $16,100 \times g$ for 30 min. The supernatant was collected as the soluble fraction. The precipitate (insoluble fraction) was washed 5-6 times with PBS and solubilized in 2% SDS containing lysis buffer by sonicating (high power, 10–15 cycles of 30 s pulses).

For the tissues (from both male and female animals of 3–5 months of age; for P1 stillborn pups, the sex was not determined), 60–70 mg of total wet mass was resuspended in the same lysis buffer, homogenized, sonicated (high power, 40 cycles of 30 s pulses), and centrifuged at $16,100 \times g$ for 30 min at 4 °C. The supernatant was collected as the soluble fraction. The precipitate (insoluble fraction) was processed as described above. Both biochemical fractions were analyzed by SDS-PAGE, and in some experiments by subsequent immunoblotting with anti-ubiquitin antibodies. Amounts of the insoluble materials were normalized and adjusted to the corresponding amounts of soluble proteins before SDS-PAGE.

## Immunoblot analyses

Lysates of cells or mouse tissues (20–100 µg) were subjected to SDS-PAGE and transferred onto a nitrocellulose membrane (GVS Life Science) with a wet blot transfer system (VWR). Membranes were blocked with 2–5% non-fat milk or BSA in TBS-Tween 20 (0.2%) and incubated with primary antibodies with the following dilutions: anti-Hsp90α (1:2000; polyclonal antiserum from Synaptic Systems and monoclonal antibodies from Enzo Lifesciences), anti-Hsp90β (1:2000), anti-Hop (1:1000), anti-GAPDH (1:7500), anti-β-actin (1:5000), anti-Hsp70 (1:2000), anti-Hsc70 (1:2000), anti-c-Raf (1:1000), anti-Ub (1:5000), anti-p23 (1:1000), anti-Cdc37 (1:1000), anti-Akt (1:1000), anti-Hsf1

(1:1000), anti-Hsp40/Hdj1 (1:1000), anti-Hsp110 (1:1000), anti-Aha1 (1:2000), anti-Hsp25/27 (1:2000), anti-Puromycin (1:22,000), anti-p-mTOR (1:1000), anti-mTOR (1:2000), anti-p-S6 (1:1000), anti-p-eIF2α (1:1000), anti-eIF2α (1:1000). Membranes were washed with TBS-Tween 20 (0.2%) and incubated with the corresponding secondary antibodies: anti-mouse IgG-HRP (1:10,000), anti-rabbit IgG-HRP (1:10,000), and anti-rat IgG-HRP (1:10,000). Immunoblots were developed using the WesternBright™ chemiluminescent substrate (Advansta). Images were recorded by using a LI-COR Odyssey or Amersham ImageQuant 800 image recorder.

## IRES and translational activity reporter assays

MAFs or RPE1 cells were seeded at a density of $3 \times 10^4$ or $6 \times 10^4$ per 0.5 ml, respectively, in 24-well cell culture plates. Cells were transfected either with plasmid pcDNA3 RLUC 5′-UTR FLUC or plasmid pcDNA3 RLUC POLIRES FLUC using PEI (1:3 DNA to PEI ratio). Plasmid pcDNA3 RLUC POLIRES FLUC was used as a positive control for IRES activity. The medium was changed 12–14 h after transfection to avoid toxicity. 48 h after transfection, cells were lysed with Passive Lysis Buffer (Promega), and firefly and Renilla luciferase activities were measured using the Dual-Luciferase detection kit (Promega) with a bioluminescence plate reader (Citation, BioTek). Firefly luciferase activities were normalized to those of Renilla luciferase, and IRES activities of the 5′-UTR were calculated by dividing the normalized firefly luciferase activities derived from the 5′-UTR reporter by those obtained with the Polio IRES for each given genotype. Similar experiments were performed under short and long-term heat shock. After transfection, MAFs or RPE1 cells were kept at 40 °C for 30 h for short-term heat shock and subsequently processed. For long-term heat stress adaptation, MAFs or RPE1 cells were maintained for 4 or 3 days at 40 °C, respectively, and subsequently transfected and processed as described above.

For translation reporter assays, MAFs were co-transfected with plasmid pGL3-CMV.Luc plasmid or its derivatives (where the firefly luciferase coding sequence is flanked by 5′- or 3′- or both UTRs of mouse *Hsp90ab1*) and a constitutive Renilla luciferase expression plasmid (pRL-CMV). Bioluminescence was detected at 48 h post-transfection, as described above. Firefly luciferase activity from pGL3-CMV.Luc transfection was used as a normalization control to determine the impact of UTRs on firefly luciferase translation. Renilla luciferase activity was used as transfection control. In all these experiments, luciferase activities were considered to be directly proportional to luciferase translation and abundance.

## RNA extraction and quantitative RT-PCR

RNA was isolated by the acid guanidinium thiocyanate-phenol-chloroform extraction method. Briefly, mouse tissues (from both male and female animals of 3–5 months of age) were homogenized, or cells were lysed, or ribosomal/polysomal precipitates were dissolved in the TRI reagent (4 M guanidium thiocyanate, 25 mM sodium citrate, 0.5% N-lauroylsarcosine, 0.1 M 2-mercaptoethanol, pH 7). Then consecutively 2 M sodium acetate pH 4, aquaphenol, and chloroform:isoamyl alcohol (49:1) were added to the lysates at a ratio of 0.1:1.0:0.2. RNA was precipitated by adding isopropanol to the aqueous phases. cDNA was prepared from RNA (400 ng) by using random primers (Promega), GoScript buffer (Promega), and reverse transcriptase (Promega) according to the manufacturer's instructions. cDNAs were mixed with the GoTaq master mix (Promega), and specific primer pairs for relevant genes (Supplementary Table 1) for quantitative PCR with Biorad CFX96 or CFX Connect thermocyclers. mRNA expression of the gene of interest was normalized to *GAPDH* or *ACTB* mRNA as internal standards.

## Phase-contrast microscopy

Cellular morphology was analyzed using an inverted light microscope (Olympus CK2) using a 5x magnification objective. Phase-contrast

images were captured with a Dino-lite camera using the DinoXcope software.

## Cell counting

Using a hemocytometer under the light microscope, we monitored cell growth and viability during heat-shock induction experiments by counting viable cells with the trypan blue exclusion assay.

## Mass spectrometry (MS)

*Mouse whole tissue MS sample preparation:* snap-frozen tissues (130–420 mg; from both male and female animals of 3–5 months of age) were resuspended in excess of chilled (−20 °C) 80% methanol and homogenized by shaking in the presence of ceramic beads on a FastPrep system for 3 cycles of 20 s pulses. After a short centrifugation, the solvent supernatants were removed, and the beads with homogenized tissues were dried. The pellets were resuspended in a lysis buffer (30 mM Tris-HCl pH 8.6, 1% Sodium deoxycholate, 10 mM DTT) in a ratio of 100 µl per 10 mg of initial tissue weight and were shaken again in the FastPrep system as described above. Then 300 µl solutions were mixed 1:1 (v/v) with lysis buffer, heated at 95 °C for 10 min, and used for all subsequent steps. Samples were digested following a modified version of the iST method[85]. Briefly, 100 µg of proteins (based on tryptophan fluorescence quantitation[86]) were diluted 1:1 (v/v) in water, chloroacetamide (at the final concentration of 32 mM) was added, and samples were incubated at 25 °C for 45 min in the dark to alkylate cysteine residues. EDTA was added to the samples at the final concentration of 3 mM, and samples were digested with 1 µg of Trypsin/LysC (Promega) for 1 h at 37 °C, followed by the identical second digestion. To remove sodium deoxycholate, two sample volumes of isopropanol containing 1% TFA were added to the digests, and the samples were desalted on a strong cation exchange (SCX) plate (Oasis MCX microelution plate; Waters Corp.) by centrifugation. Peptides were eluted in 250 µl of 80% acetonitrile, 19% water, 1% NH₃. All eluates after SCX desalting were dried, dissolved in 200 µl of 2% acetonitrile, 0.1% TFA, and 2 µl solutions were analyzed by LC-MS/MS. In the case of brain samples, peptides were separated into three fractions during SCX desalting process. Peptides were eluted with 125 mM and 500 mM ammonium acetate in 20% acetonitrile, followed by the final elution in 80% acetonitrile, 19% water, 1% NH₃. All eluates were dried and dissolved as described above, and 4 µl of samples were analyzed by LC-MS/MS.

*Whole-cell proteome MS sample preparation:* Three biological replicates of WT and Hsp90α/β KO HEK293T cells were processed as reported earlier for the WT and Hop KO HEK293T cells[38].

*General LC-MS/MS analysis:* LC-MS/MS analyses of processed samples were carried out on a Fusion Tribrid Orbitrap mass spectrometer (Thermo Fisher Scientific) interfaced through a nano-electrospray ion source to an Ultimate 3000 RSLCnano HPLC system (Dionex). Peptides were separated on a reversed-phase custom-packed 40 cm C18 column (75 µm ID, 100 Å, Reprosil Pur 1.9 µm particles; Dr. Maisch, Germany) with a 4–76% acetonitrile gradient in 0.1% formic acid (total time 140 min). Full MS survey scans were performed at 120'000 resolution. A data-dependent acquisition method controlled by Xcalibur 4.2 software (Thermo Fisher Scientific) was used that optimized the number of precursors selected ("top speed") of charge 2⁺ to 5⁺ while maintaining scan cycle time between 0.6–1.5 s. HCD fragmentation mode was used at a normalized collision energy of 32%, with a precursor isolation window of 1.6 *m/z*, and MS/MS spectra were acquired in the ion trap. Peptides selected for MS/MS were excluded from further fragmentation during 60 s.

## General data analyses

Data processing and analyses were performed using GraphPad Prism (version 8).

## Mass spectrometric data analyses

Tandem MS data of mouse tissue samples and human cell lines were processed by the MaxQuant software (version 1.6.14.0 and 1.6.3.4, respectively)[87] incorporating the Andromeda search engine[88]. The UniProt mouse reference proteome (RefProt) database of November 2019 (55,431 sequences) and human RefProt database of October 2017 (71,803 sequences) were used, supplemented with sequences of common contaminants. Trypsin (cleavage at K, R amino acid residues) was used as the enzyme definition, allowing two missed cleavages. Carbamidomethylation of cysteine was specified as a fixed modification. N-terminal acetylation of protein and oxidation of methionine were specified as variable modifications. All identifications were filtered at 1% FDR at both the peptide and protein levels with default MaxQuant parameters. iBAQ[89] values were used for quality control assessments. Protein groups labeled as reverse hits, only identified by site, and potential contaminants were removed. LFQ values[90] were further used for quantitation after log2 transformation. For the whole tissue or cell proteome experiments, from the relevant MS datasets, only protein groups were kept for the subsequent analyses that met the criteria of at least two unique peptides. For the brain dataset, we considered three unique peptides as the cut-off. In all these subsequent analyses, a log2 fold change of >0.4 or <−0.4 for a protein was considered to be a biologically significant difference, and a p-value < 0.1 was considered a statistically significant difference. Averages of biological replicates and their corresponding *p*-values were plotted as volcano plots using GraphPad Prism (version 8).

To compare the Hsp90 interactome profiles of WT and Hsp90 mutant mouse tissues, Hsp90 interactors were filtered out from the whole tissue proteomic datasets of brain (267 interactors), liver (163 interactors), and muscle (84 interactors) using data from our own web server (https://www.picard.ch/Hsp90Int/index.php, initially reported in ref. 91.). A heatmap of the calculated average fold change of the LFQ values of each genotype as compared to WT was constructed using the "expression heatmaps" tool available from: http://www.heatmapper. ca/. Fold change values were ranked with respect to the values obtained with tissues of the 90αKO 90βHET genotype.

Amounts of Hsp90α/β in proteomics samples used to compare genotypes were calculated using normalized intensities (iBAQ) as obtained from MaxQuant. Due to extensive sequence identity between the two isoforms, we verified the quantitative relationships between Hsp90α and Hsp90β using only unique (isoform-specific) peptides or "razor" and unique peptides. The presence of shared peptides did not significantly impact the overall ratios between the two isoforms (Supplementary Data 1 and 2). Peptides indicating the presence of truncated or full-length proteins of Hsp90α/β could not be unambiguously detected in KO mouse tissues, and human cells; this is consistent with the absence of such proteins in our immunoblot analyses. We therefore set the adjusted background intensities of such Hsp90α/β peptides in KO samples to 0.

## Protein and mRNA quantitation under heat shock

Mouse Hsp90α and Hsp90β mRNAs were quantified by quantitative RT-PCR, and heat shock-induced expression of these mRNAs was determined by calculating the ratio of the values obtained for 40 °C and 37 °C. Therefore, the fold change at 37 °C was set to 1. *Gapdh* was used as the reference mRNA in these experiments. Mouse Hsp90α and Hsp90β protein levels were quantified by densitometric score analyses by ImageJ-Fiji on specific immunoblot images using β actin as a loading control. Heat shock-induced changes of Hsp90α and Hsp90β protein levels were calculated similarly as done for mRNAs. Since the *Hsp90aa1* gene is mutated in mice with the 90αKO and 90αKO 90βHET genotypes, any changes of Hsp90α mRNA and protein in response to heat shock were considered insignificant and set to 0.

## Reporting summary

Further information on research design is available in the Nature Research Reporting Summary linked to this article.

## Data availability

The mass spectrometry proteomics data have been deposited to the ProteomeXchange Consortium via the PRIDE partner repository with the dataset identifier PXD031456, and for a subset they are available in Supplementary Data 1, 2. The source data are provided as Source Data files with this paper. Source data are provided with this paper.

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

## Acknowledgements
We thank Katharina Strub for her valuable advice regarding ribosome fractionation assays. We thank Matthieu Villemin for preparing the WT and Hsp90 α/β KO HEK293T cells for proteomic analyses, and Marta Madon-Simon for deleting the gene trap in the early phase of the project. We are grateful to previous other members of the Picard laboratory for miscellaneous reagents. This work has been supported by the Swiss National Science Foundation (Grant 31003A_172789/1) and the Canton de Genève.

## Author contributions
K.B. conceived the study, designed and performed experiments, analyzed the data, prepared figures, and wrote the manuscript. S.M. generated Hsp90α and Hsp90β mutant A549 cells and conducted some of the stress induction experiments with human cell lines. S.Z. and M.C. designed and performed polysome profiling experiments. L.W. and M.Q. conducted the proteomic analyses. D.H. contributed to bioinformatics analyses of the proteomic datasets. D.W. helped with mouse genotyping. L.B. contributed to generating the luciferase reporter plasmids. D.P. conceived the study, contributed to designing the experiments and analyzing the data, supervised the work, and wrote and edited the manuscript. All authors provided critical analysis of the data and contributed to the editing of the manuscript.

## Competing interests
The authors declare no competing interests.
