## [Peer Review File · Nature Communications]

Translational reprogramming in response to accumulating stressors ensures critical threshold levels of Hsp90 for mammalian lifeREVIEWER COMMENTS

Reviewer #1 (Remarks to the Author):

In this manuscript, Bhattacharya et al. employed a combination of mouse models and cultured cells to reveal the responses toward genetic depletion of HSP90 proteins. The results are supportive of a model where critical threshold levels (50-70% of normal levels) of HSP90 proteins are required for supporting mammalian life, a finding that differs substantially from what was

previously observed in yeast cells (only a few percent of normal levels is needed). In addition, low levels of HSP90 proteins were found to lead to senescence, suggesting a link between

chaperone-mediated proteostasis and aging. Overall, the findings are interesting and important to the fields of molecular chaperones and ageing research. The scholarly presentation of the

manuscript is excellent, and most experiments were carefully executed. I recommend the manuscript for publication in Nat. Commun. after some revisions:

Major Comments:

The authors observed that HSP90 β protein levels were increased in HSP90 α KO HSP90 β Het background compared to wild-type background, and the process involves an IRES present in the 5'-UTR of HSP90 β mRNA. Viewing that heat shock could induce elevated translations of HSP70 and DNAJB4 (Ref. 67 and Miao et al. Nat. Commun., 2019, 3613) through elevated levels of m6A in the 5'-UTRs of their mRNAs, the authors should examine whether a similar m6A-based pathway is at play. For instance, the authors can examine whether the 5'-UTR of HSP90 β mRNA contains a higher level of m6A in HSP90 α KO HSP90 β Het relative to WT background, and assess whether genetic depletion of METTL3, the catalytic core of the major m6A writer complex, would abolish elevated expression of HSP90 β protein in HSP90 α KO HSP90 β Het background.

I also noted that the reproducibilities for the quantitative proteomic data presented in Extended Data Tables are quite poor. Many proteins exhibit different trends in the two biological replicates.

The authors should consider adding more biological replicates to improve reliability of the proteomic data. The authors should also show scatter plots to compare the results obtained from

different replicates. In addition, it is not clear whether the data points in Volcano plots reflect the average values from the two replicates or not, and this should be noted in the figure legends.

Moreover, most significantly changed proteins should be labeled in the Volcano plots, and the authors should also label LFQ intensities as Log₂-transformed LFQ intensities.

Based on the observations that diminished levels of HSP90 proteins could contribute to elevated senescence, the authors argued that there is connection between diminished levels of HSP90 protein and aging. This is somewhat counterintuitive from the perspective that HSP90 inhibitors were shown to exert senolytic effects. Have others shown that there is a decline in HSP90 level or activity as mammals age? If not, the authors should assess whether there are age-dependent diminutions in HSP90 protein levels in mammalian tissues.

Minor corrections:

Page 6, line 132, “manages” should read as “manage”.

Page 15, line 346, delete “the”; line 361, “afore-mentioned” can be changed to “aforementioned”.

Page 19, line 445, change “thereby delay” to “thereby delaying”.

Page 23, line 557, change “thereby rejuvenate” to “thereby rejuvenating”.

Reviewer #2 (Remarks to the Author):

Review of the manuscript by Bhattacharya et al, titled “Translational reprogramming in response to accumulating stressors ensures critical threshold levels of Hsp90 for mammalian life”. In this study the authors investigate how mammals and mammalian cells ensure a critical level of the chaperone Hsp90 that is essential for viability. Using knockout of 3 out of 4 alleles of Hsp90 in mice the authors reduced the gene dosage, analyzed the surviving animals and could describe near-normal levels of Hsp90 protein. This could be explained by the identification of an internal ribosomal entry site (IRES) in the Hsp90 mRNA allowing intense translation especially under stress conditions when cap-dependent translation is repressed. Cell culture experiments defined the threshold levels of Hsp90 protein and activity that ensures cell cycle progression and viability, demonstrating that critical reduction of Hsp90 activity leads to senescence induction, proteostatic collapse and apoptosis.

This study is very well conducted with detailed and elaborated analyses leaving not much space for stimulating criticism and this is one of the rare cases where the reviewer thinks it can be accepted and published in its current form. The implications by this study for our understanding of senescence and aging justifies the publication in Nature Communications. However there are minor points that could be addressed

-lines 151-154 the sentence was not understood by the reviewer as the rectification of the Hsp90 protein levels in the KO mice does not correlate with the significant expected loss....

-lines 205-217 to strengthen the point that the translational changes after KO indicate a specific instead of a global stress response by switching from cap-dependent to IRES translation (as indicated by the high p-eIF2a), the authors could analyze the ribosome-bound mRNA by sequencing and IRES analysis or alternatively perform analyses with cap/IRES reporters in the KO cells. In my view the selective probing of other

chaperones/co-chaperones under stress does not justify the speculations in the result part. The introduction of ITAFs as a mechanistical explanation for the observed IRES-dependent rectification of Hsp90 protein in the discussion is a very good point.

- Figure 3b. Here the reviewer is not convinced that it is ok to plot together different time points in one bar to get statistics....

-Figure 5f y axis labeling is different to 5D

-Figure 5h why is a wt control missing?

Reviewer #3 (Remarks to the Author):

In this manuscript, Bhattacharya et al reported that mammalian Hsp90 needs to maintain certain levels for the organismal survival. In a mouse strain lacking Hsp90aa1 but with a single Hsp90ab1 allele, the survived mice had to increase the Hsp90ab1 expression. The authors proposed that this adaptative Hsp90ab1 induction occurred at the translational level via IRES. Finally, the authors conclude that below threshold levels of Hsp90 leads to proteotoxic stress and senescence.

Using genetic mouse models and cells in culture, the manuscript contains comprehensive data sets covering proteomics and reporter assays. The results clearly indicate that, unlike other species, mammals need a much higher threshold level of Hsp90. My biggest concern is the underling mechanism of Hsp90ab1 adaptation. The authors appeared to ignore the increased Hsp90ab1 mRNAs levels in survived animals. This is very much like genetic compensation at the transcriptional level. Although it is possible that the 5'UTR of Hsp90ab1 bears the IRES activity, it is premature to exclude the transcriptional adaptation. In fact, the changed mRNA levels might contribute more to the protein abundance, considering the possibility that mRNA is generally short-lived. Overall, a large portion of the presented data in this manuscript is informative, but the claimed translational reprogramming of Hsp90ab1 is premature, if not misleading.

Major concerns:

1. Figure 3, it is unclear whether the UTR reporter assays also measured the reporter mRNA levels in parallel. This is important because some UTR sequence features might influence the mRNA stability.

2. Figure 3f, why the 5'UTR of Hsp90ab1 only responds in 90(alpha)KO90(beta)HET cells? Typical IRES responds to a wide range of stress conditions. It is very surprising that the IRES is inactive in 90(alpha)KO cells. This again speaks to the possibility of transcriptional adaptation of Hsp90ab1.
3. The IRES of Hsp90ab1 needs to be better characterized. If the secondary structure is difficult to determine, at least some inactive mutants are required to confirm the IRES feature.
4. It is also puzzling that the Hsp90ab1 adaptation is not a common feature because only a few mice managed to survive by achieving this. The authors need to comment on this, at least in the discussion section.
5. Under Hsp90ab1 adaptation, Hsp90 levels were back to normal, and the animal survived. But why the global protein synthesis was still severely reduced as shown in Figure 3? Does this contribute to the delayed aging phenotype?

Point-by-point response to reviewers' comments

Note that our responses to the reviewers' comments are in blue. The manuscript has undergone some formatting changes to comply with the style of Nature Communications. Moreover, we would also like to point out that some of the figure and reference numbers have changed due to insertions.

We very much appreciate the reviewers' critical reading and constructive suggestions.

REVIEWERS' COMMENTS

Reviewer #1 (Remarks to the Author):

In this manuscript, Bhattacharya et al. employed a combination of mouse models and cultured cells to reveal the responses toward genetic depletion of HSP90 proteins. The results are supportive of a model where critical threshold levels (50-70% of normal levels) of HSP90 proteins are required for supporting mammalian life, a finding that differs substantially from what was previously observed in yeast cells (only a few percent of normal levels is needed). In addition, low levels of HSP90 proteins were found to lead to senescence, suggesting a link between chaperone-mediated proteostasis and aging. Overall, the findings are interesting and important to the fields of molecular chaperones and ageing research. The scholarly presentation of the manuscript is excellent, and most experiments were carefully executed. I recommend the manuscript for publication in Nat. Commun. after some revisions:

Major Comments:

The authors observed that HSP90 β protein levels were increased in HSP90 α KO HSP90 β Het background compared to wild-type background, and the process involves an IRES present in the 5'-UTR of HSP90 β mRNA. Viewing that heat shock could induce elevated translations of HSP70 and DNAJB4 (Ref. 67 and Miao et al. Nat. Commun., 2019, 3613) through elevated levels of m6A in the 5'-UTRs of their mRNAs, the authors should examine whether a similar m6A-based pathway is at play. For instance, the authors can examine whether the 5'-UTR of HSP90 β mRNA contains a higher level of m6A in HSP90 α KO HSP90 β Het relative to WT background, and assess whether genetic depletion of METTL3, the catalytic core of the major m6A writer complex, would abolish elevated expression of HSP90 β protein in HSP90 α KO HSP90 β Het background.

We already had a whole paragraph about these aspects in the Discussion. We have now added DNAJB4 as an example (and the corresponding reference). However, exploring this further goes beyond the scope of this study, notably also because the mechanisms of how stress leads to more m6A in the 5' UTRs is still controversial.

I also noted that the reproducibilities for the quantitative proteomic data presented in Extended Data Tables are quite poor. Many proteins exhibit different trends in the two biological replicates.

The authors should consider adding more biological replicates to improve reliability of the proteomic data. The authors should also show scatter plots to compare the results obtained from different replicates. In addition, it is not clear whether the data points in Volcano plots reflect the average values from the two replicates or not, and this should be noted in the figure legends.

Moreover, most significantly changed proteins should be labeled in the Volcano plots, and the authors should also label LFQ intensities as Log2-transformed LFQ intensities.

As suggested, we have compared the corresponding two replicates with scatter plots and calculated the Pearson correlation coefficients (new panel b of Supplementary Fig. 2 and corresponding text; note that the numbering for all subsequent Supplementary Figures had to be shifted). This analysis shows that the correlation is close to 1 in all pairs. We would also like to point out that we deliberately used tissues from one male and one female (as was mentioned in the Results), which may also account for some (slight) differences between the proteomes of the two replicates.

The data points in Volcano plots are always average values of the replicates. Although already implicit in the Methods and all legends, we have now added an explicit statement to the Methods.

As suggested, we have added the names of the most strongly changed proteins in the Volcano plots of Supplementary Fig. 3 (originally Extended Data Fig. 2). All labels of the X-axes of the Volcano plots already mentioned that log₂ values are plotted.

Based on the observations that diminished levels of HSP90 proteins could contribute to elevated senescence, the authors argued that there is connection between diminished levels of HSP90 protein and aging. This is somewhat counterintuitive from the perspective that HSP90 inhibitors were shown to exert senolytic effects. Have others shown that there is a decline in HSP90 level or activity as mammals age? If not, the authors should assess whether there are age-dependent diminutions in HSP90 protein levels in mammalian tissues.

We already had a whole paragraph about this (the senolytic effects of Hsp90 inhibitors and their apparently contradictory effects) in the Discussion (the last one of the Discussion), and we also explicitly mentioned the correlation between aging and reduced Hsp90 levels in the Abstract. We took advantage of this revision to add two additional references in the Results.

Minor corrections:

Page 6, line 132, “manages” should read as “manage”. *fixed*

Page 15, line 346, delete “the”; *added another "the"*

line 361, “afore-mentioned” can be changed to “aforementioned”. *fixed*

Page 19, line 445, change “thereby delay” to “thereby delaying”. *fixed*

Page 23, line 557, change “thereby rejuvenate” to “thereby rejuvenating”. *fixed*

Reviewer #2 (Remarks to the Author):

Review of the manuscript by Bhattacharya et al, titled "Translational reprogramming in response to accumulating stressors ensures critical threshold levels of Hsp90 for mammalian life". In this study the authors investigate how mammals and mammalian cells ensure a critical level of the chaperone Hsp90 that is essential for viability. Using knockout of 3 out of 4 alleles of Hsp90 in mice the authors reduced the gene dosage, analyzed the surviving animals and could describe near-normal levels of Hsp90 protein. This could be explained by the identification of an internal ribosomal entry site (IRES) in the Hsp90 mRNA allowing intense translation especially under stress conditions when cap-dependent translation is repressed. Cell culture experiments defined the threshold levels of Hsp90 protein and activity that ensures cell cycle progression and viability, demonstrating that critical reduction of Hsp90 activity leads to senescence induction, proteostatic collapse and apoptosis. This study is very well conducted with detailed and elaborated analyses leaving not much space for stimulating criticism and this is one of the rare cases where the reviewer thinks it can be accepted and published in its current form. The implications by this study for our understanding of senescence and aging justifies the publication in Nature Communications. However there are minor points that could be addressed

-lines 151-154 the sentence was not understood by the reviewer as the rectification of the Hsp90 protein levels in the KO mice does not correlate with the significant expected loss....

We have now completely rewritten this sentence (and broken it up into several sentences) and hope that it is clearer now.

-lines 205-217 to strengthen the point that the translational changes after KO indicate a specific instead of a global stress response by switching from cap-dependent to IRES translation (as indicated by the high p-eIF2a), the authors could analyze the ribo-seq mRNA by sequencing and IRES analysis or alternatively perform analyses with cap/IRES reporters in the KO cells. In my view the selective probing of other chaperones/co-chaperones under stress does not justify the speculations in the result part. The introduction of ITAFs as a mechanistical explanation for the observed IRES-dependent rectification of Hsp90 protein in the discussion is a very good point.

Since sequencing the ribo-seq mRNA would be a major enterprise and beyond the scope of this study, we would like to point out to the reviewer that the luciferase assays of Fig. 3e and f, as pointed out in the Methods, were done with a standard cap-dependent internal transfection control ("Renilla").

Regarding the comment: "In my view the selective probing of other chaperones/co-chaperones under stress does not justify the speculations in the result part", we are a bit at a loss to understand what is meant. Our probing was far from selective. By Western, we looked at a whole range of chaperones/co-chaperones and by proteomics at everything that wasn't there only in minute amounts. There is no way we could have missed a global stress response. And then, not all stresses are alike. This is not a heat shock experiment, but a genetic model where proteostasis is

challenged by initially reduced levels of the most abundant molecular chaperone (Hsp90).

- Figure 3b. Here the reviewer is not convinced that it is ok to plot together different time points in one bar to get statistics....

Good point, thank you for spotting this oversight. Indeed, the data we have are 2 time points with 2 replicates each. We decided to highlight the two time points within the existing Fig. 3b, and to be as clear as possible in the legend. We now state explicitly that the trend is the same if we plot the two time points separately, and that we calculated the p-values solely as an indication, and how exactly we did that.

-Figure 5f y axis labeling is different to 5D

Must be, because one is mouse (f) and the other human (d).

-Figure 5h why is a wt control missing?

We didn't include wt in this analysis, because its response was already known and because the level of senescence in wt is low. Moreover, we do not compare the different genotypes, but for each genotype minus and plus rapamycin. The primary aim of this experiment was to check whether rapamycin is an efficient "therapeutic" agent in the Hsp90 deficiency-driven aging models.

Reviewer #3 (Remarks to the Author):

In this manuscript, Bhattacharya et al reported that mammalian Hsp90 needs to maintain certain levels for the organismal survival. In a mouse strain lacking Hsp90aa1 but with a single Hsp90ab1 allele, the survived mice had to increase the Hsp90ab1 expression. The authors proposed that this adaptative Hsp90ab1 induction occurred at the translational level via IRES. Finally, the authors conclude that below threshold levels of Hsp90 leads to proteotoxic stress and senescence.

Using genetic mouse models and cells in culture, the manuscript contains comprehensive data sets covering proteomics and reporter assays. The results clearly indicate that, unlike other species, mammals need a much higher threshold level of Hsp90. My biggest concern is the underlying mechanism of Hsp90ab1 adaptation. The authors appeared to ignore the increased Hsp90ab1 mRNAs levels in survived animals. This is very much like genetic compensation at the transcriptional level. Although it is possible that the 5'UTR of Hsp90ab1 bears the IRES activity, it is premature to exclude the transcriptional adaptation. In fact, the changed mRNA levels might contribute more to the protein abundance, considering the possibility that mRNA is generally short-lived. Overall, a large portion of the presented data in this manuscript is informative, but the claimed translational reprogramming of Hsp90ab1 is premature, if not misleading.

We would like to point out to the reviewer that we extensively looked at endogenous Hsp90 mRNA levels (and corresponding protein levels) and presented the data across multiple figures. All our data, including the newly added ones, unambiguously

show that it is translational reprogramming, and not mRNA levels, which accounts for the increased levels of Hsp90 β .

This being an important issue, we appreciate the reviewer's next comment (Major concerns #1), which raises this point in a different context, too.

Major concerns:

1. Figure 3, it is unclear whether the UTR reporter assays also measured the reporter mRNA levels in parallel. This is important because some UTR sequence features might influence the mRNA stability.

Very well taken. We have now done this experiment and include the data for the mRNA analysis in Supplementary Fig. 7c, d, f. It strongly supports our conclusion that Hsp90 β protein levels in the context of the 90 α KO 90 β HET genotype are most likely due to increased translation driven by IRES activity of the 5'UTR of the Hsp90 β mRNA.

2. Figure 3f, why the 5'UTR of Hsp90ab1 only responds in 90(alpha)KO90(beta)HET cells? Typical IRES responds to a wide range of stress conditions. It is very surprising that the IRES is inactive in 90(alpha)KO cells. This again speaks to the possibility of transcriptional adaptation of Hsp90ab1.

Only the 90 α KO 90 β HET cells are "equipped" to reprogram Hsp90 β translation by exploiting the IRES activity of the 5'UTR of the Hsp90 β mRNA. Other genotypes essentially need both UTRs to reach maximal translation (as known for conventional cap-dependent translation), as stated in the corresponding Results section.

We respectfully disagree with the reviewer's comment that a typical cellular IRES must respond equally to a wide range of stresses. The subset of known mRNAs utilizing their IRES varies dramatically depending on the stress, e.g. heat stress, hypoxia, mitosis, apoptosis, or differentiation. Each and every Hsp90 mutant that we have looked at may generate a somewhat different (genetically caused) stress. Thus, the Hsp90 β IRES is specifically active in 90 α KO 90 β HET cells. A positive value (more than 0) in our experiments with bicistronic reporters (Fig. 3e; Supplementary Fig. 7g) means there is IRES activity, and clearly it is highest in the 90 α KO 90 β HET background. We now clarify what the Y-axis means in the legend to Fig. 3.

Moreover, we were not surprised not to see strong IRES activity in 90 α KO compared to 90 α KO 90 β HET cells since similar steady-state Hsp90 β protein levels are found (Fig. 2c) while 90 α KO cells both have a much higher rate of global translation (Fig. 3a) and an increased Hsp90 β mRNA level (Supplementary Fig. 5b).

3. The IRES of Hsp90ab1 needs to be better characterized. If the secondary structure is difficult to determine, at least some inactive mutants are required to confirm the IRES feature.

While this is an interesting challenge for future research, this is an entire project by itself. Cellular IRES have not been as extensively characterized as viral ones, and each IRES has its own specifics (including ITAFs and m6A, as already discussed in our manuscript). With the current knowledge it is very difficult to predict what type of mutations should be made to disrupt the IRES. While we cannot say how that IRES works, all our data taken together (the activity of bicistronic reporters, the comparison of endogenous Hsp90 β protein and mRNA levels, in normal and heat-stressed conditions) strongly support the existence of an IRES activity in the Hsp90 β mRNA.

4. It is also puzzling that the Hsp90ab1 adaptation is not a common feature because only a few mice managed to survive by achieving this. The authors need to comment on this, at least in the discussion section.

This is the whole point, and that's why we did discuss it extensively in the Discussion (3rd paragraph from the end).

5. Under Hsp90ab1 adaptation, Hsp90 levels were back to normal, and the animal survived. But why the global protein synthesis was still severely reduced as shown in Figure 3? Does this contribute to the delayed aging phenotype?

The global cap-dependent translation is strongly reduced because of the inactivity of mTORC1 and hyperphosphorylation of eIF2 α in 90 α KO 90 β HET (Supplementary Fig. 7a) even when Hsp90 β levels are increased. Therefore, the translational reprogramming is specific for the Hsp90 β mRNA as an adaptive mechanism and unrelated to effects on global translation.

We had already mentioned in the Results section (last sentence of the first chapter) that the lifespan of the survivors appears to be similar to wild-type. These survivors are inherently adapted and their lifespan (an aging phenotype) may essentially not be altered under normal living conditions. However, we hypothesized that reduced global translation together with rectified (overexpressed) Hsp90 β levels could lead to a favorable chaperone to substrate ratio (also discussed in the Discussion section). One could also speculate that this could contribute to a delayed aging phenotype in 90 α KO β HET under stressed conditions. To avoid ethically critical stress experiments with mice, we did such experiments with the MAFs using long-term mild heat stress as a mediator of accelerated senescence, and the findings fully support our hypothesis (Fig. 5f,g; Supplementary Fig. 12a-d). However, how these 90 α KO 90 β HET mouse survivors cope with such a low level of global translation is an exciting question for future research.

REVIEWERS' COMMENTS

Reviewer #1 (Remarks to the Author):

The authors have satisfactorily addressed most of the comments raised by this reviewer, and the revised manuscript is significantly improved. While I still think that examining whether an m6A-based epitranscriptomic mechanism contributes to the IRES-mediated up-regulation of HSP90beta would enhance the study, I recognized that a substantial body of work has been presented in this manuscript. Hence, it is OK to save the subject for a future study, and I support the acceptance of this nice piece of work for publication in Nat. Commun. after a few very minor corrections:

Abstract, line 5, the sentence can be rephrased to "...four alleles of genes encoding cytosolic Hsp90, with one HSP90beta allele...".

Page 21, 5th line from bottom, "base" can be changed to "basal".

Page 24, line 11, "thereafter" perhaps can be changed to "above threshold levels".

Reviewer #2 (Remarks to the Author):

The authors did address satisfactorily the minor points that were raised by me in the first round of revision. I recommend the acceptance of the manuscript without further revision.

Reviewer #3 (Remarks to the Author):

In this revised manuscript, the authors conducted some experiments to address my previous concerns, which are centered on mRNA steady state levels. This is important to exclude the transcriptional adaptation. I am not entirely convinced by the authors' explanation that why there is no IRES activity in 90αKO compared to 90αKO 90βHET cells. Why did 90αKO cells have a much higher rate of global translation (Fig. 3a) and an increased Hsp90β mRNA level (Supplementary Fig. 5b)? The authors' rebuttal

letter is brief, which is also true for responses to other Reviewers' comments. Since the manuscript is potentially interesting and informative, a better job could have been done to make the revised manuscript more suitable for publication.

Point-by-point response to reviewers' comments

Note that our responses to the reviewers' comments are in blue. The manuscript has undergone some formatting changes to comply with the style of Nature Communications. We very much appreciate the reviewers' renewed critical reading and constructive suggestions, which have once again allowed us to improve the manuscript further.

REVIEWERS' COMMENTS

Reviewer #1 (Remarks to the Author):

The authors have satisfactorily addressed most of the comments raised by this reviewer, and the revised manuscript is significantly improved. While I still think that examining whether an m6A-based epitranscriptomic mechanism contributes to the IRES-mediated up-regulation of HSP90beta would enhance the study, I recognized that a substantial body of work has been presented in this manuscript. Hence, it is OK to save the subject for a future study, and I support the acceptance of this nice piece of work for publication in Nat. Commun. after a few very minor corrections:

Abstract, line 5, the sentence can be rephrased to "...four alleles of genes encoding cytosolic Hsp90, with one HSP90beta allele...".

Thanks for the suggestion. We modified it.

Page 21, 5th line from bottom, "base" can be changed to "basal".

We changed it.

Page 24, line 11, "thereafter" perhaps can be changed to "above threshold levels".

Must have been left over from previous editing. We adjusted the sentence.

Reviewer #2 (Remarks to the Author):

The authors did address satisfactorily the minor points that were raised by me in the first round of revision. I recommend the acceptance of the manuscript without further revision.

Reviewer #3 (Remarks to the Author):

In this revised manuscript, the authors conducted some experiments to address my previous concerns, which are centered on mRNA steady state levels. This is important to exclude the transcriptional adaptation. I am not entirely convinced by the authors' explanation that why there is no IRES activity in 90αKO compared to 90αKO 90βHET cells. Why did 90αKO cells have a much higher rate of global translation (Fig. 3a) and an increased Hsp90β mRNA level (Supplementary Fig. 5b)? The authors' rebuttal

letter is brief, which is also true for responses to other Reviewers' comments. Since the manuscript is potentially interesting and informative, a better job could have been done to make the revised manuscript more suitable for publication.

We apologize if our revised manuscript and the response to this reviewer's comment was not sufficiently clear. Regarding the manuscript, we have now edited the relevant paragraph to state certain arguments more explicitly (and more succinctly than in the following).

What is really key is that the increased global translation of 90 α KO cells is not generally true. In brain, for example, it is not the case. In 90 α KO MAFs, which display this substantial increase, this might be specific to this particular cell/tissue type and/or a clonal effect. The changes of global translation and the increased Hsp90 β mRNA levels in 90 α KO MAFs are not the focus of this discussion, and indeed, figuring this out might be a whole project in itself. What is relevant here is the relative translation of Hsp90 β . To be able to compare Hsp90 β translation across different genotypes, we have to standardize it to global translation. More specifically and importantly, 90 α KO 90 β HET cells produce similar amounts of Hsp90 β protein as 90 α KO cells despite reduced global translation, and therefore, the unaltered Hsp90 β mRNA levels strongly suggest translational reprogramming. The result is unambiguous. Throughout several different kinds of experiments, there is always a relative increase of Hsp90 β translation.